# Aligning Text to Image in Diffusion Models is Easier Than You Think

**Jaa-Yeon Lee**[*1]     **Byunghee Cha**[*1]     **Jeongsol Kim**[2]     **Jong Chul Ye**[1]

[1]Kim Jaechul Graduate School of AI, KAIST
[2]Department of Bio and Brain Engineering, KAIST

{jaayeon, paulcha1025, jeongsol, jong.ye}@kaist.ac.kr

[*]Equal contribution

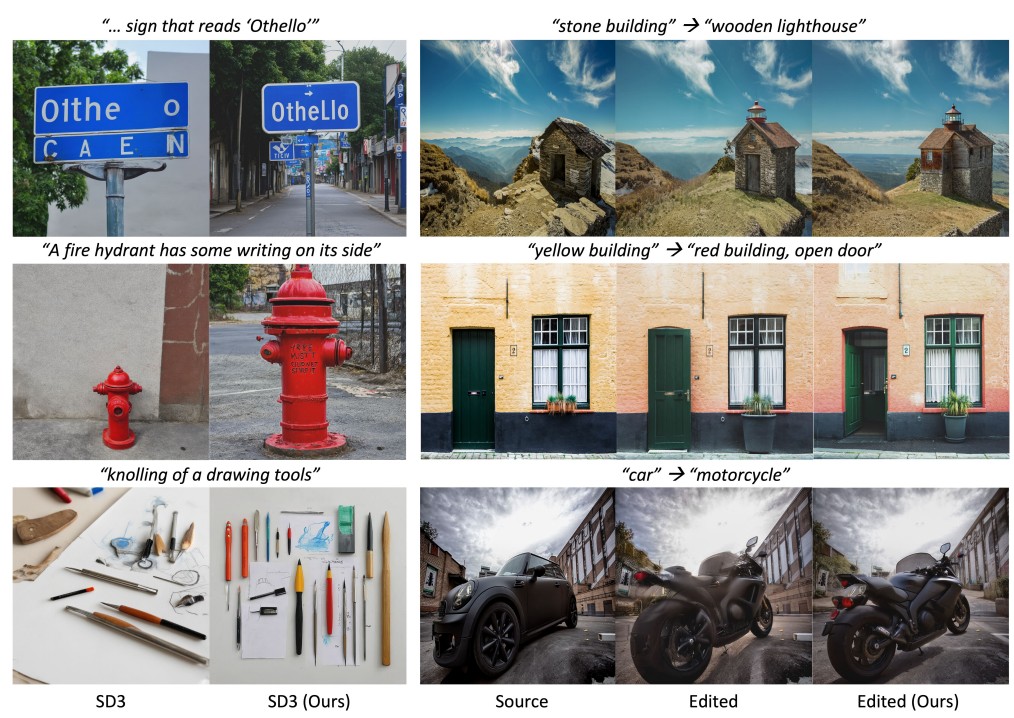

*"… sign that reads 'Othello'"*      *"stone building" → "wooden lighthouse"*

*"A fire hydrant has some writing on its side"*      *"yellow building" → "red building, open door"*

*"knolling of a drawing tools"*      *"car" → "motorcycle"*

SD3      SD3 (Ours)           Source      Edited      Edited (Ours)

Figure 1: **Representative results for image generation and image editing**. SoftREPA provides much improved text-to-image alignment by introducing a negligible size of learnable soft tokens.

## Abstract

While recent advancements in generative modeling have significantly improved text-image alignment, some residual misalignment between text and image representations still remains. Some approaches address this issue by fine-tuning models in terms of preference optimization, etc., which require tailored datasets. Orthogonal to these methods, we revisit the challenge from the perspective of representation alignment—an approach that has gained popularity with the success of REPresentation Alignment (REPA) [46]. We first argue that conventional text-to-image (T2I) diffusion models, typically trained on paired image and text data (i.e., positive pairs) by minimizing score matching or flow matching losses, is suboptimal from the standpoint of representation alignment. Instead, a better

39th Conference on Neural Information Processing Systems (NeurIPS 2025).

alignment can be achieved through contrastive learning that leverages existing dataset as both positive and negative pairs. To enable efficient alignment with pretrained models, we propose *SoftREPA*—a lightweight contrastive fine-tuning strategy that leverages soft text tokens for representation alignment. This approach improves alignment with minimal computational overhead by adding fewer than 1M trainable parameters to the pretrained model. Our theoretical analysis demonstrates that our method explicitly increases the mutual information between text and image representations, leading to enhanced semantic consistency. Experimental results across text-to-image generation and text-guided image editing tasks validate the effectiveness of our approach in improving the semantic consistency of T2I generative models. Project Page: `https://softrepa.github.io/`.

# 1 Introduction

Achieving effective alignment between modalities is essential in multimodal generative modeling. Specifically, latent diffusion models [33] enable various conditioning mechanisms, such as text, during the generation process for the semantic alignment between the two modalities. For UNet-based denoisers [33, 29, 31], image representations are updated to align with fixed text representations encoded by pre-trained CLIP models through cross-attention layers. In contrast, transformer-based denoisers [30, 8] jointly update both text and image representations by concatenating them and processing them through self-attention layers. However, there is still room for further improvement, and to mitigate the remaining misalignment between text and image representations, a more effective representation learning approach is necessary.

Recent advancements in representation alignment for Diffusion Transformers (DiT) [30] have notably enhanced their ability to learn semantically meaningful internal representations [46, 11]. Notably, REPA [46] showed that aligning the internal representations of DiT with an external pre-trained visual encoder during training significantly improves both discriminative and generative performance. To further enhance text-image alignment in text-to-image (T2I) generative models, we leverage these ideas for aligning DiT's internal representations.

Most vision-language generative foundation models have focused on improving multimodal alignment through architectural modifications. These include MLP-based projection layers (e.g., LLaVA-style visual instruction tuning) [25, 26], cross-attention-based fusion mechanisms (e.g., Flamingo, Stable Diffusion 1.5) [2**?** ], and early fusion models that integrate text and vision features within a shared representation space (e.g., Chameleon, Stable Diffusion 3, FLUX) [37, 8]. In fact, many of these approaches have been largely adopted in T2I generative models. However, we explore an efficient yet effective way of further enhancing the representation alignment for given pre-trained models.

Specifically, we focus on the contrastive learning framework, a widely adopted strategy for multimodal alignment in vision-language representation learning, as seen in models such as CLIP [32], ALBEF [21], BLIP [22], and ALIGN [12]. One of the key contributions of this work is the introduction of *soft text tokens*, which are optimized via contrastive image-text training. These soft tokens allow the model to dynamically adapt its text representations, improving alignment with generated images without requiring full model fine-tuning. This simple yet effective approach significantly enhances text-to-image alignment in both text-to-image generation and text-guided image editing tasks. Notably, the method is highly flexible and can be seamlessly integrated with any pretrained text-to-image (T2I) generative model. From a theoretical standpoint, we show that this approach explicitly increases the mutual information between text and image representations, resulting in improved semantic consistency across modalities. We refer to our method as *SoftREPA*, short for Soft REPresentation Alignment. Our contributions can be summarized as follows.

- We propose SoftREPA, a novel text-image representation alignment method that leverages a lightweight fine-tuning strategy with soft text tokens. This approach improves text-image alignment while adding fewer than 1M additional parameters, ensuring efficiency with minimal computational overhead.

- SoftREPA is simple yet flexible so that it can be used with any pretrained T2I generative models to improve performance of image generation, editing, etc.

- We show that our method explicitly increases mutual information between image and text, leading to better semantic consistency in multi-modal representations.

## 2 Preliminaries

**Flow Models** Suppose that we have access to samples from target distribution $X_0 \sim q$ and source distribution $X_1 \sim p$. The goal of flow model is to generate $X_0$ starting from $X_1$. Specifically, we define a velocity field $v_t(\boldsymbol{x})$ of a flow $\psi_t(\boldsymbol{x}) : [0, 1] \times \mathbb{R}^d \to \mathbb{R}^d$ that satisfies $\psi_t(X_0) = X_t$ and $\psi_1(X_0) = X_1$. Here, the $\psi_t$ is uniquely characterized by a flow ODE:

$$d\psi_t(\boldsymbol{x}) = v_t(\psi_t(\boldsymbol{x}))dt \tag{1}$$

where the flow velocity $v_t$ is fitted to the parameterized neural network $v_{t,\theta}$ via flow matching:

$$\mathcal{L}_{FM} = \mathbb{E}_{t \in [0,1], \boldsymbol{x}_t \sim p_t} \|v_t(\boldsymbol{x}_t) - v_{t,\theta}(\boldsymbol{x}_t)\|^2. \tag{2}$$

However, this is computationally expensive to solve due to the integration with respect to $X_0$, so the authors in [24] proposed a conditional flow matching that has the same gradient with the original objective function:

$$\mathcal{L}_{CFM} = \mathbb{E}_{t \in [0,1], \boldsymbol{x}_0 \sim q} \|v_t(\boldsymbol{x}_t|\boldsymbol{x}_0) - v_{t,\theta}(\boldsymbol{x}_t)\|^2, \tag{3}$$

where $v_t(\boldsymbol{x}_t|\boldsymbol{x}_0)$ defines a conditional flow $\psi_t(\boldsymbol{x}_t|\boldsymbol{x}_0)$ satisfying $\psi_t(\boldsymbol{x}_1|\boldsymbol{x}_0) = \boldsymbol{x}_t$. In particular, the linear conditional flow defines the flow as $\boldsymbol{x}_t = \psi_t(\boldsymbol{x}_1|\boldsymbol{x}_0) = (1-t)\boldsymbol{x}_0 + t\boldsymbol{x}_1$. Then, the conditional velocity field is given by $v_t(\boldsymbol{x}_t|\boldsymbol{x}_0) = \dot{\psi}_t(\psi_t^{-1}(\boldsymbol{x}_t|\boldsymbol{x}_0)|\boldsymbol{x}_0) = \boldsymbol{x}_1 - \boldsymbol{x}_0$ [24]. Thus, the conditional flow matching loss is defined as

$$\mathbb{E}_{t \in [0,1], \boldsymbol{x}_0, \boldsymbol{x}_1 \sim \pi_{0,1}} \|(\boldsymbol{x}_1 - \boldsymbol{x}_0) - v_{t,\theta}(\boldsymbol{x}_t)\|^2. \tag{4}$$

Considering a marginal velocity field,

$$
\begin{aligned}
v_t(\boldsymbol{x}_t) &= \int v_t(\boldsymbol{x}_t|\boldsymbol{x}_0)p(\boldsymbol{x}_0|\boldsymbol{x}_t)d\boldsymbol{x}_0 = \mathbb{E}[v_t(\boldsymbol{x}_t|\boldsymbol{x}_0)|\boldsymbol{x}_t] \\
&= \mathbb{E}[\boldsymbol{x}_1 - \boldsymbol{x}_0|\boldsymbol{x}_t] = \mathbb{E}[\boldsymbol{x}_1|\boldsymbol{x}_t] - \mathbb{E}[\boldsymbol{x}_0|\boldsymbol{x}_t],
\end{aligned}
\tag{5}
$$

the generation process via flow ODE is

$$d\boldsymbol{x}_t = v_t(\boldsymbol{x}_t)dt = (\mathbb{E}[\boldsymbol{x}_1|\boldsymbol{x}_t] - \mathbb{E}[\boldsymbol{x}_0|\boldsymbol{x}_t])dt. \tag{6}$$

Accordingly,

$$d\boldsymbol{x}_t = \left( \mathbb{E}\left[ \frac{\boldsymbol{x}_t - (1-t)\boldsymbol{x}_0}{t}|\boldsymbol{x}_t \right] - \mathbb{E}[\boldsymbol{x}_0|\boldsymbol{x}_t] \right) dt = \frac{\boldsymbol{x}_t - \mathbb{E}[\boldsymbol{x}_0|\boldsymbol{x}_t]}{t}dt \tag{7}$$

which is the probability-flow ODE (PF-ODE) of DDIM [35, 15] when $\boldsymbol{x}_1 \sim p := \mathcal{N}(0, \boldsymbol{I}_d)$. In other words, flow models and score-based models can be used interchangeably.

**Contrastive Representation Learning** Contrastive learning aims to maximize the similarity between semantically related text-image pairs while pushing apart unrelated pairs in a shared representation space [6, 32]. Formally, given a batch of $N$ image-text pairs $(\boldsymbol{I}_i, \boldsymbol{T}_i)_{i=1}^N$, contrastive learning optimizes a contrastive loss function, typically a variant of the InfoNCE loss:

$$\mathcal{L}_{\text{CLIP}} = -\frac{1}{N} \sum_{i=1}^N \log \frac{\exp(\text{sim}(\boldsymbol{I}_i, \boldsymbol{T}_i)/\tau)}{\sum_{j=1}^N \exp(\text{sim}(\boldsymbol{I}_i, \boldsymbol{T}_j)/\tau)} \tag{8}$$

where $\text{sim}(\boldsymbol{I}_i, \boldsymbol{T}_i)$ is a similarity metric such as cosine similarity or inner product between image and text embeddings, and $\tau$ denotes a temperature parameter that controls the distribution sharpness. This contrastive loss encourages image and text representations to form a joint multimodal representation space, where aligned pairs are close together, and unaligned pairs are separated. Several fundamental models have successfully applied contrastive learning for multi-modal representation learning [32, 12, 21, 22] and self-supervised learning [6].

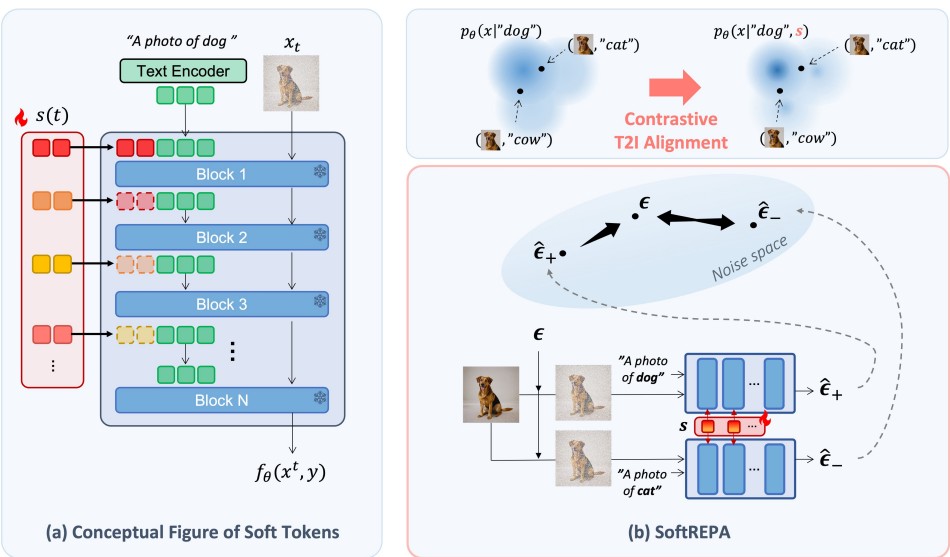

Figure 2: **Network architecture and algorithmic concept of SoftREPA.** (a) Learnable soft tokens of each layer are prepended to the text features across the upper layers. (b) The soft tokens are optimized to contrastively match the score with positively conditioned predicted noise while repelling the score from negatively conditioned predicted noise. This process implicitly sharpens the joint probability distribution of images and text by reducing the log probability of negatively paired conditions.

Despite the success of contrastive learning in representation learning, its application in generative models remains underexplored due to key challenges. First, there is a fundamental mismatch between discriminative and generative features; conventional contrastive learning optimizes a representation space for discriminative tasks, whereas generative models focus on realistic sample synthesis. Second, text-image alignment in diffusion-based generative models is often implicit, relying on denoising objectives rather than explicit representation learning.

To bridge this gap, we propose a contrastive learning framework that aligns both representation learning and generative objectives, ensuring effective text-image representation alignment while preserving generative quality.

## 3  SoftREPA

### 3.1  Contrastive T2I Alignment Loss

Let $\{(\boldsymbol{x}^{(i)}, \boldsymbol{y}^{(i)})\}_{i=1}^{n}$ denotes the matched image and text pairs as a training data set. Then, contrastive text-to-image alignment in T2I model can be generally formulated as follows:

$$\mathcal{L} = -\frac{1}{n} \sum_{i=1}^{n} \log \frac{\exp(l(\boldsymbol{x}^{(i)}, \boldsymbol{y}^{(i)}))}{\sum_j \exp(l(\boldsymbol{x}^{(i)}, \boldsymbol{y}^{(j)}))} \tag{9}$$

where $l(\cdot, \cdot)$ represents the similarity measure. It is important to note that unlike the standard T2I model training that only consider positive pairs, i.e. $(\boldsymbol{x}^{(i)}, \boldsymbol{y}^{(i)})$, our SoftREPA employs the contrastive T2I alignment that additionally considers negative image and text pairs $(\boldsymbol{x}^{(i)}, \boldsymbol{y}^{(j)})), i \neq j$. This significantly improves the alignment performance.

Furthermore, we define $l(\boldsymbol{x}^{(i)}, \boldsymbol{y}^{(j)})$ as the logit from the variation of the denoising score matching loss for the case of T2I diffusion models [41].

$$l(\boldsymbol{x}, \boldsymbol{y}) = e^{-\mathbb{E}_{t,\boldsymbol{\epsilon}}[\|\boldsymbol{\epsilon}_\theta(\boldsymbol{x}_t, t, \boldsymbol{y}) - \boldsymbol{\epsilon}\|^2 / \tau(t)]} \tag{10}$$

Similarly, in conditional flow matching, the logit $l(\boldsymbol{x}, \boldsymbol{y})$ can be formalized as:

$$l(\boldsymbol{x}, \boldsymbol{y}) = e^{-\mathbb{E}_{t,\boldsymbol{\epsilon}}[\|v_\theta(\boldsymbol{x}_t, t, \boldsymbol{y}) - (\boldsymbol{\epsilon} - \boldsymbol{x}_0)\|^2 / \tau(t)]} \tag{11}$$

Here, $\tau(t)$ represents time scheduling parameter. One might consider using the denoising score matching loss, $-\mathbb{E}_{t,\epsilon}[\|\epsilon_\theta(\boldsymbol{x}_t, t, \boldsymbol{y}) - \epsilon\|^2]$, as a similarity measure instead of relying on its logit value. However, our experiments revealed that this formulation can introduce instability during training due to the unbounded nature of the similarity measure. To address this, SoftREPA introduce an exponential function to constrain the logit values, effectively stabilizing the training process.

Intuitively, as illustrated in Fig. 2(b), training a diffusion model with this loss encourage the predicted noise to align with the true noise when conditioned on a matching text description. Conversely, when conditioned on a mismatched text, the predicted noise deviates further from the true noise. This sharpens the conditional probability distribution of the trained model and increases the distinction between different text conditions, which can enhance image-text alignment in generated images.

## 3.2 The Soft Tokens

To distill the contrastive score matching objective above, we introduce a learnable soft token $\boldsymbol{s}$, which is the only trainable parameter while pretrained model remains frozen. In particular, as illustrated in fig. 2, we introduce a set of learnable soft tokens that vary between layers and are indexed by time $t$. The soft token at layer $k$ is defined using an embedding function with the number of soft tokens as $m$:

$$\mathbf{s}^{(k,t)} = \text{Embedding}(k, t) \in \mathbb{R}^{m \times d} \tag{12}$$

At the layer $k$, the text representation is updated by concatenating the time-indexed soft token with text hidden representation from the previous layer: $\hat{\boldsymbol{H}}_{\text{text}}^{(k-1,t)} = [\mathbf{s}^{(k,t)}; \boldsymbol{H}_{\text{text}}^{(k-1,t)}] \in \mathbb{R}^{(m+n) \times d}$. The soft token is prepended on text features across the layers, leading to a modified denoising function: $v_\theta(\boldsymbol{x}_t, t, \boldsymbol{y}, \boldsymbol{s})$. By adopting the soft tokens, we can adjust the image and text representations to get better semantic alignment. Additionally, for computational efficiency, we approximate the expectation within $l(\boldsymbol{x}, \boldsymbol{y})$ using a single Monte Carlo sample, effectively replacing it as

$$\tilde{l}(\boldsymbol{x}, \boldsymbol{y}, \boldsymbol{s}) = e^{-\|v_\theta(\boldsymbol{x}_t, t, \boldsymbol{y}, \boldsymbol{s}) - (\epsilon - \boldsymbol{x}_0)\|^2 / \tau(t)} \tag{13}$$

which can stabilize the computation using the same $\epsilon$ and $t$ in the same batch. The resulting contrastive loss function with soft tokens can be defined as follows:

$$\mathcal{L}_{\text{SoftREPA}}(\boldsymbol{s}) = -\mathbb{E}_{(\boldsymbol{x}, \boldsymbol{y}) \sim p_{\text{data}}, t \sim U(0,1), \epsilon \sim \mathcal{N}(\mathbf{0}, \mathbf{I})} \log \left( \frac{\exp(\tilde{l}(\boldsymbol{x}, \boldsymbol{y}, \boldsymbol{s}))}{\sum_j \exp(\tilde{l}(\boldsymbol{x}, \boldsymbol{y}^{(j)}, \boldsymbol{s}))} \right) \tag{14}$$

Then, the learnable token $\boldsymbol{s}$ is optimized by minimizing $\mathcal{L}_{\text{SoftREPA}}(\boldsymbol{s})$. During inference, the trained soft tokens are leveraged with fixed text representation from the text encoder across layers and timesteps. The detailed procedure during the image generation process based on MM-DiT architecture is described in algorithm 1 from appendix B.

## 3.3 Relationship with Mutual Information

Although the relationship between contrastive loss and mutual information has been previously studied, we revisit it here to provide a more complete theoretical perspective. Additionally, we emphasize the critical role of the logit formulation in the score matching loss, highlighting its importance in ensuring stable and effective optimization.

According to [18], pointwise mutual information (PMI) over image $\boldsymbol{x}$ and text $\boldsymbol{y}$ pairs can be defined as follows:

$$i(\boldsymbol{x}, \boldsymbol{y}) = \log \frac{p_\theta(\boldsymbol{x}|\boldsymbol{y})}{p_\theta(\boldsymbol{x})} = \log \frac{p_\theta(\boldsymbol{x}|\boldsymbol{y})}{\mathbb{E}_{p(\boldsymbol{c})}[p_\theta(\boldsymbol{x}|\boldsymbol{c})]} \tag{15}$$

where the second equality follows from the definition of conditional expectation. Song et al. [36] and Kong et al. [18] further show that assuming the diffusion model is an optimal denoiser, the conditional likelihood $p_\theta(\boldsymbol{x}|\boldsymbol{y})$ can be described as (see appendix A):

$$p_\theta(\boldsymbol{x}|\boldsymbol{y}) = \exp(\hat{l}(\boldsymbol{x}, \boldsymbol{y})) \quad \text{where} \quad \hat{l}(\boldsymbol{x}, \boldsymbol{y}) = -\frac{1}{2} \int_0^T \lambda(t) \mathbb{E}[\|\epsilon_\theta(\boldsymbol{x}_t, t, \boldsymbol{y}) - \epsilon\|^2] dt + C. \tag{16}$$

By approximating the denominator of PMI via Monte Carlo sampling, we arrive at:

$$i(\boldsymbol{x}, \boldsymbol{y}) \approx \log \frac{\exp(\hat{l}(\boldsymbol{x}, \boldsymbol{y}))}{\frac{1}{N} \sum_{\boldsymbol{c}} \exp(\hat{l}(\boldsymbol{x}, \boldsymbol{c}))} \tag{17}$$

| | COCO val5K | | | | | | Efficiency | |
| | Human Preference | | Text Alignment | | Image Quality | | PM | Latency |
| Model | ImageReward↑ | PickScore↑ | CLIP↑ | HPS↑ | FID↓ | LPIPS↓ | (GB) | (sec/image) |
|---|---|---|---|---|---|---|---|---|
| SD1.5 | 17.72 | 21.47 | 26.4 | 25.08 | 24.59 | 43.80 | 2.621 | 1.526 |
| SD1.5 +Ours | **32.89** | **21.50** | **27.33** | **25.18** | **23.43** | **43.38** | 2.621 | 1.547 |
| SDXL | 75.06 | 22.38 | 26.76 | 27.35 | **24.69** | **42.05** | 10.45 | 4.059 |
| SDXL +Ours | **85.29** | **22.62** | **26.80** | **28.30** | 26.04 | 42.39 | 10.45 | 4.060 |
| SD3 | 94.27 | 22.54 | 26.30 | 28.09 | **31.59** | **42.43** | 18.77 | 4.637 |
| SD3 +Ours | **108.5** | **22.55** | **26.91** | **28.91** | 36.21 | 42.88 | 18.77 | 4.695 |

| | GenEval | | | | | | |
| Model | Mean↑ | Single↑ | Two↑ | Counting↑ | Colors↑ | Position↑ | Color Attribution↑ |
|---|---|---|---|---|---|---|---|
| SD3 | 0.68 | 0.99 | 0.86 | 0.56 | 0.85 | 0.27 | 0.55 |
| CaPO [20] | 0.71 | 0.99 | 0.87 | 0.63 | 0.86 | 0.31 | 0.59 |
| RankDPO [16] | **0.74** | **1.00** | 0.90 | **0.72** | 0.87 | 0.31 | 0.66 |
| SD3 + Ours | 0.70 | **1.00** | **0.95** | 0.29 | **0.92** | **0.34** | **0.68** |

Table 1: Quantitative evaluation of T2I generation with learnable soft tokens on SD1.5, SDXL, and SD3. Generation quality is evaluated on the COCO-val 5K [23] and GenEval [9] benchmark. Peak GPU memory(PM) usage and per-image latency are measured to assess computational efficiency. ImageReward, CLIP, HPS, and LPIPS are scaled by $\times 10^2$.

This formulation closely resembles a cross-entropy loss, where the conditional log-likelihood $\hat{l}(\boldsymbol{x}, \boldsymbol{y})$ can be interpreted as logits. Now, from the fact that mutual information is the expectation of the pointwise mutual information, we have

$$I(X, Y) = \frac{1}{n} \sum_{i=1}^{n} \log \frac{\exp(\hat{l}(\boldsymbol{x}^{(i)}, \boldsymbol{y}^{(i)}))}{\sum_j \exp(\hat{l}(\boldsymbol{x}^{(i)}, \boldsymbol{y}^{(j)}))} + D \tag{18}$$

where $\boldsymbol{x}^{(i)}$ represents the $i$-th sample in the data, $n$ denotes the number of samples in the data, and $D$ indicates the constant. Notably, Eq. (18) is very similar in form to the contrastive objective in Eq. (9), where the logit is the negative of the diffusion loss. This implies that minimizing our contrastive learning objective is closely related to maximizing the mutual information between image-text pairs under the diffusion model.

## 4 Experiments

**Implementation details.** In our experiments, to confirm the flexibility of SoftREPA, we utilize various open-source T2I diffusion models, including Stable Diffusion 1.5, Stable Diffusion XL, and Stable Diffusion 3 to evaluate text and image alignment in both image generation and text-based image editing tasks. For the training of softs token, the batch size was set to 16 with less than 30,000 iterations using two A100 GPUs. For Stable Diffusion 3, we set the length of the soft token to 4 and used 5 layers to attach the soft tokens for further experiments. In the case of Stable Diffusion 1.5 and Stable Diffusion XL, soft tokens are applied only to the Down or Middle block layers of the UNet. Refer to further implementation details in appendix B.

**Text to Image Generation** We conducted text-to-image generation experiments on SD1.5, SDXL, and SD3, training soft tokens using the text-image paired COCO dataset [23]. The implementation details are provided in appendix B. To evaluate the generated images, we assessed human preference scores [45, 17], text-image alignment [48, 44], and image quality [47, 34] on COCO-val 5K dataset. Furthermore, we conducted evaluation on GenEval [9], compared with RankDPO [16], which is fine-tuned via preference optimization. As shown in table 1, in COCO dataset, the proposed method generally outperforms baseline approaches in both diffusion and rectified flow models regardless of the model architecture. Additionally, in GenEval, SD3 with SoftREPA mostly outperforms RankDPO [16]. The drop in the counting metric can be primarily attributed to the tendency of diffusion models to interpret text alignment in a way that encourages the generation of multiple instances of the referenced object. Further analysis and potential mitigation strategies are discussed in appendix D. In fig. 3, detailed textual descriptions are more accurately reflected in the generated images, as demonstrated in the qualitative results.

| | Inversion | Method | Human Preference | | Text Alignmnet | | | Structure | Background Preservation | | |
|---|---|---|---|---|---|---|---|---|---|---|---|
| | | | Image-Reward↑ | Pick-Score↑ | CLIP/Edited↑ | CLIP/Whole↑ | HPS↑ | Distance↓ | PSNR↑ | LPIPS/Whole↓ | SSIM↑ |
| PIEBench | ddim | PnP | 32.29 | 21.53 | 22.56 | 25.59 | 26.11 | **27.32** | **22.31** | 14.48 | **79.58** |
| | ddim | PnP (Ours) | **36.78** | **21.59** | **22.70** | **25.78** | **26.35** | 27.43 | 22.27 | 14.55 | 79.36 |
| | direct | PnP | 40.85 | 21.65 | 22.68 | 25.64 | 26.60 | **23.33** | 22.46 | **13.44** | **80.22** |
| | direct | PnP (Ours) | **43.32** | **21.70** | **22.80** | **25.83** | **26.76** | 23.41 | 22.40 | 13.53 | 79.93 |
| | ddim | MasaCtrl | -13.94 | 21.03 | 21.20 | 24.18 | 23.59 | 27.63 | 22.31 | 14.59 | **80.41** |
| | ddim | MasaCtrl (Ours) | **-12.76** | **21.05** | **21.27** | **24.44** | **23.69** | **27.36** | 22.27 | 14.5 | 80.27 |
| | direct | MasaCtrl | **4.77** | 21.39 | 21.47 | 24.54 | 24.92 | 23.61 | **22.82** | 12.21 | **82.02** |
| | direct | MasaCtrl (Ours) | 4.48 | **21.40** | **21.49** | **24.69** | **24.93** | **23.32** | 22.77 | **12.19** | 81.85 |
| | - | FlowEdit | 87.70 | 22.16 | 23.19 | 26.72 | 28.04 | 25.48 | 24.37 | 12.55 | 88.58 |
| | - | FlowEdit (Ours) | **102.24** | **22.31** | **23.60** | **27.19** | **28.58** | **24.07** | **24.99** | 12.60 | **88.73** |
| DIV2K | ddim | PnP | -9.72 | 21.10 | - | 26.07 | 24.76 | 32.39 | - | 18.89 | - |
| | ddim | PnP (Ours) | **-7.75** | **21.15** | - | **26.18** | **24.88** | **31.95** | - | **18.88** | - |
| | direct | PnP | -6.86 | 21.17 | - | 25.96 | 25.09 | 29.15 | - | **17.96** | - |
| | direct | PnP (Ours) | **-5.56** | **21.22** | - | **26.15** | **25.17** | **28.82** | - | 17.99 | - |
| | ddim | MasaCtrl | -60.30 | 20.55 | - | 23.37 | 21.44 | 29.70 | - | 17.40 | - |
| | ddim | MasaCtrl (Ours) | **-57.41** | **20.59** | - | **23.46** | **21.64** | **28.79** | - | **17.03** | - |
| | direct | MasaCtrl | -42.33 | 20.80 | - | 23.65 | 22.7 | 71.16 | - | 31.41 | - |
| | direct | MasaCtrl (Ours) | **-41.20** | **20.81** | - | **23.68** | **22.77** | **71.11** | - | **31.26** | - |
| | - | FlowEdit | 38.08 | 21.74 | - | 26.05 | 25.53 | 39.68 | - | 15.48 | - |
| | - | FlowEdit (Ours) | **46.68** | **21.88** | - | **26.39** | **26.03** | **35.66** | - | **14.98** | - |
| C2D | - | FlowEdit | 93.70 | 20.72 | - | 22.51 | 26.68 | **33.41** | - | **19.92** | - |
| | - | FlowEdit (Ours) | **114.4** | **21.22** | - | **23.43** | **28.35** | 34.05 | - | 20.47 | - |

Table 2: Quantitative evaluation of image editing performance with the use of soft tokens on PnP [38], MasaCtrl [4], Direct inversion [14], and FlowEdit [19]. FlowEdit is based on SD3, and others are based on SD1.5 architecture. The editing methods are evaluated on PIEBench [14], DIV2K [1], and Cat2Dog(C2D). ImageReward, CLIP, HPS, LPIPS, and SSIM are scaled by $\times 10^2$ and Distance is scaled by $\times 10^3$.

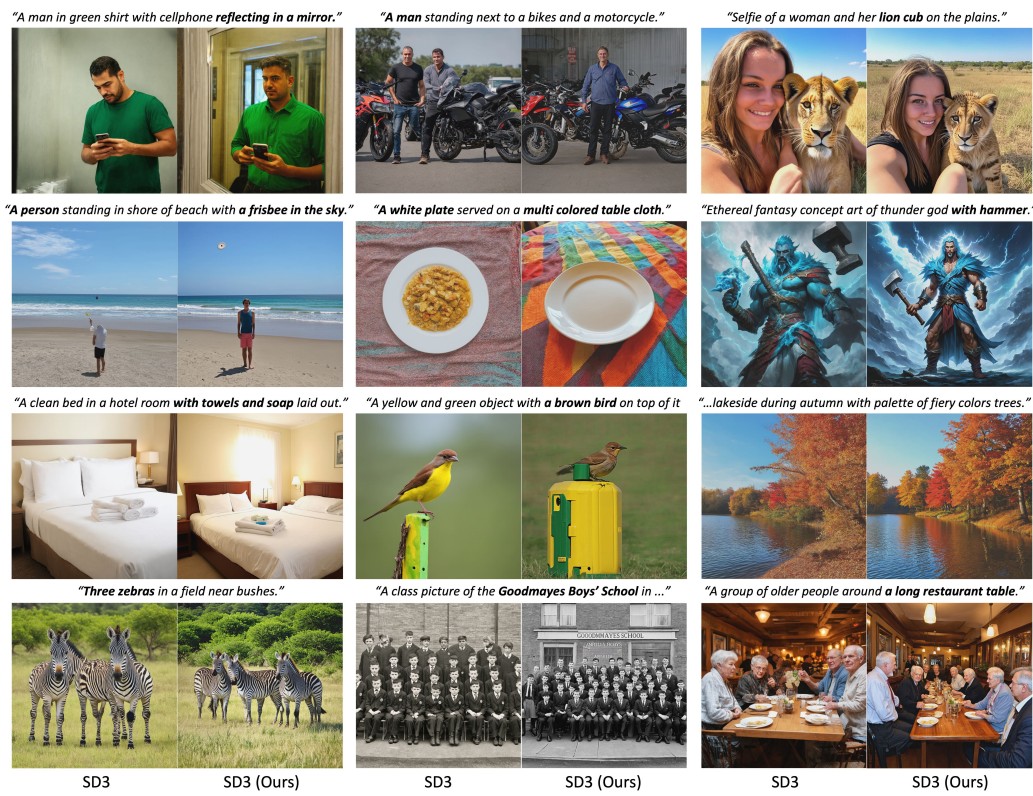

*"A man in green shirt with cellphone **reflecting in a mirror**."*  *"**A man** standing next to a bikes and a motorcycle."*  *"Selfie of a woman and her **lion cub** on the plains."*

*"**A person** standing in shore of beach with **a frisbee in the sky**."*  *"**A white plate** served on a **multi colored table cloth**."*  *"Ethereal fantasy concept art of thunder god **with hammer**."*

*"A clean bed in a hotel room **with towels and soap** laid out."*  *"A yellow and green object with **a brown bird** on top of it*  *"...lakeside during autumn with palette of fiery colors trees."*

*"**Three zebras** in a field near bushes."*  *"A class picture of the **Goodmayes Boys' School** in ..."*  *"A group of older people around **a long restaurant table**."*

SD3    SD3 (Ours)        SD3    SD3 (Ours)        SD3    SD3 (Ours)

Figure 3: The qualitative results of text-to-image generation comparing SD3 and SD3 with proposed method. The given text is from COCO and Pixart dataset.

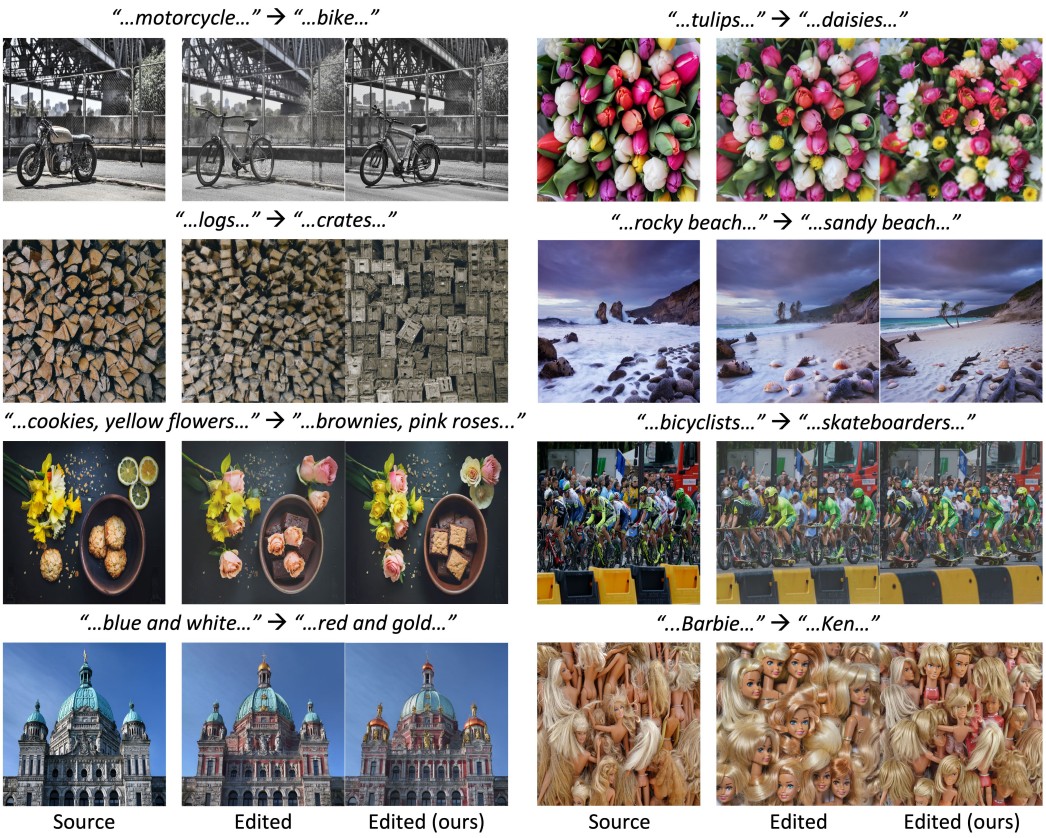

*"…motorcycle…"* → *"…bike…"*    *"…tulips…"* → *"…daisies…"*

*"…logs…"* → *"…crates…"*    *"…rocky beach…"* → *"…sandy beach…"*

*"…cookies, yellow flowers…"* → *"…brownies, pink roses…"*    *"…bicyclists…"* → *"…skateboarders…"*

*"…blue and white…"* → *"…red and gold…"*    *"…Barbie…"* → *"…Ken…"*

Source        Edited        Edited (ours)        Source        Edited        Edited (ours)

Figure 4: The qualitative results of text guided image editing comparing on SD3 with the proposed method. The FlowEdit [19] is used as the editing method for both baseline and SoftREPA.

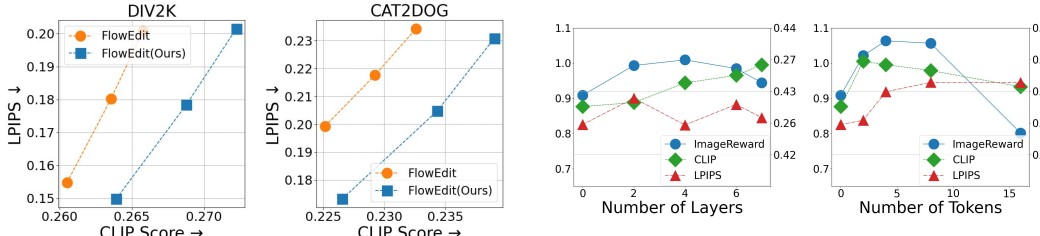

Figure 5: Quantitative comparison of baseline and SoftREPA on DIV2K and Cat2Dog editing using CLIP Score and LPIPS with various CFG scales.

Figure 6: Ablation study on the number of layers and token length for soft tokens. Metrics are normalized to: ImageReward $[0.7, 1.1]$, CLIP score $[0.25, 0.28]$, and LPIPS $[0.42, 0.44]$.

**Memory and Speed Comparison**    In table 1, we report the peak GPU memory usage (PM) and per-image latency for models with and without soft tokens, including SD1.5, SDXL, and SD3. All models were evaluated in float16 precision. The use of soft tokens introduces no change in peak memory and results in only a negligible increase in latency.

**Text Guided Image Editing**    To further demonstrate the improved text-image alignment achieved by our method, we conducted text-guided image editing experiments using PnP [38], MasaCtrl [4] with DDIM [35], Direct Inversion [14], and FlowEdit [19]. FlowEdit employs SD3, while the other methods use SD1.5. For evaluation, we used PIEBench [14], DIV2K [1], and Cat2Dog datasets. For DIV2K editing dataset, we selected 800 high-quality images and generated detailed source and target captions using LLaVA [25]; dataset construction details are provided in appendix C. In table 2,

| | Human Preference | | Text Alignment | | Image Quality | |
|---|---|---|---|---|---|---|
| Model | ImageReward↑ | PickScore↑ | CLIP↑ | HPS↑ | FID↓ | LPIPS↓ |
| SD3 | 90.89 | 22.50 | 26.26 | 27.97 | **56.87** | **42.47** |
| SD3 + LoRA + Contrastive loss | 97.72 | **22.57** | 26.42 | 28.44 | 70.33 | 42.71 |
| SD3 + Soft tokens + Contrastive loss (Ours) | **106.32** | 22.49 | **26.92** | **28.70** | 60.76 | 42.99 |

Table 3: Quantitative comparison of T2I generation (COCO-val 1K [23]) between soft tokens and LoRA on SD3. ImageReward, CLIP, HPS, and LPIPS are scaled by $\times 10^2$.

the quantitative results of human preference, text alignment, and structure distance show consistent enhancements with the use of soft tokens without compromising structure and background fidelity.

Considering that FlowEdit performs text-based image editing, where performance heavily depends on text-image alignment, this result indicates improved efficiency in modeling the joint text-image distribution using the proposed method. Furthermore, our method with soft tokens achieves more efficient editing by reducing the number of editing steps to 30, compared to 33 in FlowEdit. As shown in fig. 5, incorporating soft tokens across various CFG scales consistently enhances text-image alignment and image quality. Figure 4 further demonstrates that performance has been improved on both single and multi-concept editing compared to vanilla SD3. Additional results and details are provided in appendix F.

**Ablation Study**    We conducted an ablation study on SD3 using the COCO val dataset [23] to evaluate the effects of the number of layers using soft text tokens and token length on generation quality (fig. 6). A value of 0 corresponds to vanilla SD3, and the number of layers is counted from the first. Comparisons were made using either a fixed token length of 8 or a fixed depth of 5 layers. Our findings indicate that incorporating soft tokens beyond layer 7 severely degrades image generation quality. The optimal performance, aligned with human preferences (ImageReward [45]), was observed when soft tokens were applied within layers $2 \sim 5$. This aligns with the architecture, where early layers handle text-image alignment and later ones enhance fidelity [27]. Notably, CLIP scores continued to improve as more layers used soft tokens, indicating stronger adherence to text prompts, while overall image quality remained stable. Supporting qualitative results are provided in appendix G. For token length, using more than 8 tokens negatively impacted quality, suggesting overfitting to the fine-tuning data. Shorter lengths (1–4 tokens) preserved perceptual similarity (LPIPS), while moderate lengths (4–8 tokens) significantly improved text-image alignment (CLIP and ImageReward). Qualitative examples are shown in appendix G.

**Comparison with LoRA**    To further assess the effectiveness of our design, we examined whether contrastive learning alone could achieve comparable improvements. Specifically, we conducted an ablation study replacing the soft tokens with LoRA. As shown in table 3, the combination of soft tokens and contrastive loss consistently outperforms all other variants across evaluation metrics. This result demonstrates the strong synergy between soft tokens and contrastive learning in enhancing text–image alignment. These findings validate the core motivation of SoftREPA—to explore the text feature space for improved alignment between textual and visual representations. Additional comparisons with different LoRA ranks and layer configurations are presented in appendix H.

**Complementarity with Diffusion RL methods**    We examined the complementarity between SoftREPA and diffusion reinforcement learning (RL) methods. Since these approaches are not mutually exclusive, we tested whether SoftREPA can enhance diffusion RL frameworks. Specifically, we integrated pretrained soft tokens from SoftREPA into two representative methods—Diffusion-DPO [43], which leverages preference optimization, and DDPO [3], which applies policy optimization with an external reward—using a plug-and-play approach without additional training. As shown in table 4, incorporating SoftREPA's soft tokens consistently improves performance. This suggests strong complementarity while diffusion RL methods align outputs via preference or reward signals, SoftREPA refines internal text–image representations through contrastive learning. Combining the two effectively unites explicit preference alignment with robust representation learning to enhance generation quality.

| Method | Human Preference | | Text Alignment | | Image Quality | |
|---|---|---|---|---|---|---|
| | ImageReward↑ | PickScore↑ | CLIP↑ | HPS↑ | FID↓ | LPIPS↓ |
| Diffusion-DPO | 22.26 | **21.68** | 26.51 | **25.60** | 54.19 | 43.96 |
| Diffusion-DPO + Ours | **33.10** | 21.64 | **27.41** | 25.55 | **53.72** | **43.64** |
| DDPO | -8.83 | 21.20 | 26.21 | 23.45 | 54.83 | 43.85 |
| DDPO + Ours | **8.07** | **21.25** | **27.13** | **23.75** | **54.18** | **43.59** |

Table 4: Complementarity of SoftREPA with diffusion RL methods (Diffusion-DPO [43], DDPO [3]). Soft tokens are integrated in a plug-and-play manner during inference. Generation quality is evaluated on the COCO-val 1K dataset [23]. ImageReward, CLIP, HPS, and LPIPS are scaled by $\times 10^2$.

# 5 Related Works

Several methods have been proposed to address the text-image misalignment. Training free approach such as CFG++ [7] addresses this problem as the off-manifold phenomenon of conventional CFG and reformulates it to a manifold-constrained CFG to get better text-to-image alignment. Other training-free approach, attend-and-excite [5], adjust attention activations to enhance text-image consistency. DOODL [42] applies guidance to end-to-end manner. Meanwhile, training-based approaches such as Diffusion-DPO [43] introduce alignment losses during training to both enforce stronger text-conditioning and visual realism. Following Diffusion-DPO, denoising diffusion policy optimization(DDPO) [3] enhances the alignment using feedback from the pre-trained vision-language model. RankDPO [16] presents fully synthetic, ranking-based preference datasets generated by reward models instead of humans. However, these methods require either extensive full-finetuning or an additional annotation dataset. In this paper, we use the prior knowledge of diffusion model itself to get better text and image alignment, proposing a novel SoftREPA on soft tokens.

To enable efficient prompt-level fine-tuning, VPT [13] introduced learnable tokens for vision transformers while keeping the backbone frozen. CoOp [50] and CoCoOp [49] proposed prompt learning methods for CLIP-like vision-language models by optimizing prompts to distinguish between a predefined set of classes in image classification tasks. In recent works, MinorityPrompt [39] and Optical-Prompt [28] learned sample-specific prompts to improve diversity and temporal coherence during sampling steps, respectively. In contrast to these approaches, which typically tailor prompts to individual samples or specific class sets, we propose learning general-purpose, sample-independent soft tokens. These tokens are optimized over a distribution of text-image pairs, $(\boldsymbol{X}, \boldsymbol{Y})$, effectively guiding the propagation of text features to better align with the joint text-image distribution.

# 6 Conclusion

In this work, we introduce SoftREPA, a novel approach for improving text-image alignment in generative models. Building on the success of prior representation alignment techniques, we extend the idea of internal representation alignment to the text-image alignment without relying on external encoders. Our method effectively integrates contrastive learning and score matching loss to enhance text-image representation learning while preserving the generative capabilities of diffusion models. Also, we introduced soft text tokens, which can effectively adjust the image-text representations through contrastive T2I alignment, making it lightweight and efficient. We demonstrate that our approach enhances text-image alignment in both image generation and editing tasks while maintaining a minimal parameter overhead. Our experiments span various models, including Stable Diffusion 1.5, XL, and 3.

**Limitation and Potential Negative Impacts.** While soft tokens effectively enhance text-image alignment, they may risk overemphasizing textual guidance at the expense of faithfully preserving the user's intent in the prompt. Future work could explore regularization techniques to mitigate this issue. Additionally, this work does not address potential risks such as data poisoning or adversarial attacks during soft token training. Although these concerns are beyond the scope of this paper, they merit consideration in future research. Moreover, our method inherits any potential negative impacts of the underlying backbone models.

## Acknowledgments and Disclosure of Funding

This work was supported by the Institute of Information & communications Technology Planning & Evaluation (IITP) grant funded by the Korean government(MSIT) (No. RS-2024-00457882, AI Research Hub Project). This work was supported by the National Research Foundation of Korea under Grant RS-2024-00336454.

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

# SoftREPA for better Text-to-Image Alignment

## Supplementary Materials

## A  Information Theoretical Analysis of Diffusion Models

Several studies have explored diffusion models from an information-theoretic viewpoint. According to [18], the mutual information between random variables $X$ and $Y$ can be expressed as the expectation of pointwise mutual information over their joint distribution:

$$I(X, Y) = \mathbb{E}_{p(\boldsymbol{x}, \boldsymbol{y})}[i(\boldsymbol{x}, \boldsymbol{y})], \tag{19}$$

where the pointwise mutual information (PMI) is defined as the difference between the conditional and marginal log-likelihoods of a given sample $\boldsymbol{x}$:

$$i(\boldsymbol{x}, \boldsymbol{y}) = \log p(\boldsymbol{x}|\boldsymbol{y}) - \log p(\boldsymbol{x}). \tag{20}$$

This quantity measures how much the presence of condition $\boldsymbol{y}$ affects the probability distribution near $\boldsymbol{x}$.

Song et al. [36] and Kong et al. [18] showed that, under the assumption of an optimal diffusion model trained on given data, the log-likelihood of an image $\boldsymbol{x}$ can be formulated as:

$$-\log p(\boldsymbol{x}) = \frac{1}{2} \int_0^T \lambda(t) \mathbb{E}_{\boldsymbol{\epsilon}}[\|\boldsymbol{\epsilon}_\theta(\boldsymbol{x}_t, t) - \boldsymbol{\epsilon}\|^2] dt + C \tag{21}$$

This result also extends to conditional diffusion models, where the likelihood of $\boldsymbol{x}$ given condition $\boldsymbol{y}$ (e.g., text embeddings) is given by:

$$-\log p(\boldsymbol{x}|\boldsymbol{y}) = \frac{1}{2} \int_0^T \lambda(t) \mathbb{E}_{\boldsymbol{\epsilon}}[\|\boldsymbol{\epsilon}_\theta(\boldsymbol{x}_t, t, \boldsymbol{y}) - \boldsymbol{\epsilon}\|^2] dt + C \tag{22}$$

## B  Implementation Details

**Architecture Details**  For Stable Diffusion 3, the main experiments use soft tokens of length 4, applied across 5 layers. In Stable Diffusion 1.5, soft tokens of length 4 are applied only to the Down block layers of the UNet's conditional text embeddings. For Stable Diffusion XL, soft tokens of length 8 are used, applied to both the Down and Middle block layers of the UNet's conditional text embeddings, without incorporating timestep dependency.

**Training Details**  The hyperparameters used during training are listed in table 5. For Stable Diffusion 3, the soft tokens are optimized solely using the contrastive score matching loss ($L_{\text{SoftREPA}}$). In contrast, for Stable Diffusion 1.5 and Stable Diffusion XL, optimization combines the contrastive score matching loss ($L_{\text{SoftREPA}}$) with a small weighting of the denoising score matching loss ($L_{\text{DSM}}$).table 6 shows that the model consistently achieves improved performance regardless of whether the denoising score matching loss is applied. We observed that initializing the soft tokens with a random distribution led to performance degradation in Stable Diffusion 1.5. To address this, we initialized the soft tokens using unconditional text embeddings. The code is publicly available at `https://github.com/softrepa/SoftREPA`.

| Models | lr | wd | batch size (positive, negative) | iterations | token init | optimizer | lr scheduler |
|--------|-----|-----|------------------------------|-----------|-----------|-----------|--------------|
| SD1.5 | 1e-3 | 1e-4 | 32(4, 28) | 26,000 | $\varnothing$ | AdamW | CosineAnnealingWarmRestarts |
| SDXL | 1e-3 | 1e-4 | 16(1, 15) | 30,000 | $N(0, 0.02)$ | AdamW | CosineAnnealingWarmRestarts |
| SD3 | 1e-3 | 1e-4 | 16(4, 12) | 30,000 | $N(0, 0.02)$ | AdamW | CosineAnnealingWarmRestarts |

Table 5: The implementation details for training.

| Model | loss | Human Preference | | Text Alignment | | Image Quality | |
|---|---|---|---|---|---|---|---|
| | | ImageReward↑ | PickScore↑ | CLIP↑ | HPS↑ | FID↓ | LPIPS↓ |
| SD1.5 | - | 17.72 | 21.47 | 26.45 | 25.08 | 24.59 | 43.80 |
| | $L_{\text{SoftREPA}} + L_{\text{DSM}}$ | **32.89** | 21.50 | **27.33** | 25.18 | **23.43** | **43.38** |
| | $L_{\text{SoftREPA}}$ | 32.63 | **21.61** | 27.10 | **25.94** | 29.25 | 44.09 |
| SDXL | - | 75.06 | 22.38 | 26.76 | 27.35 | **24.69** | **42.05** |
| | $L_{\text{SoftREPA}} + L_{\text{DSM}}$ | **85.29** | **22.62** | 26.80 | 28.30 | 26.04 | 42.39 |
| | $L_{\text{SoftREPA}}$ | 81.09 | 22.59 | **26.87** | **28.32** | 26.42 | 42.69 |

Table 6: Quantitative comparison of T2I generation between contrastive score matching loss ($L_{\text{SoftREPA}}$) with and without denoising score matching loss ($L_{\text{DSM}}$). Generation quality is evaluated on the COCO-val 1K [23]. ImageReward, CLIP, HPS, and LPIPS are scaled by $\times 10^2$.

---

**Algorithm 1** Image Generation with Soft Tokens in MM-DiT

---

**Require:** Gaussian noise $\mathbf{z} \sim \mathcal{N}(\mathbf{0}, \mathbf{I})$
**Require:** Text $\boldsymbol{Y} \sim p_{\text{data}}$
**Require:** Soft token $\mathbf{s} \sim \text{Embedding}(k, t)$
**Require:** Number of layers $N$, Threshold layer $L$
**Require:** Time steps $\{t_T, t_{T-1}, ..., t_0\}$
1: Initialize $\boldsymbol{H}_{\text{img}}^{(0,T)} \leftarrow \mathbf{z}$
2: Initialize $\boldsymbol{H}_{\text{text}}^{(0,T)} \leftarrow \text{TextEncoder}(\boldsymbol{Y})$
3: $n = |\boldsymbol{H}_{\text{text}}^{(0,T)}|$
4: **for** $t$ in $\{t_T, t_{T-1}, ..., t_0\}$ **do**
5:     **for** $l = 1$ to $N$ **do**
6:         **if** $k \leq L$ **then**
7:             $\mathbf{s}^{(k,t)} \leftarrow \text{Embedding}(k, t)$
8:             $\hat{\boldsymbol{H}}_{\text{text}}^{(k-1,t)} \leftarrow [\mathbf{s}^{(k,t)}; \boldsymbol{H}_{\text{text}}^{(k-1,t)}]$
9:         **else**
10:             $\hat{\boldsymbol{H}}_{\text{text}}^{(k-1,t)} \leftarrow \boldsymbol{H}_{\text{text}}^{(k-1,t)}$
11:         **end if**
12:         $\boldsymbol{H}_{\text{img}}^{(k,t)}, \hat{\boldsymbol{H}}_{\text{text}}^{(k,t)} \leftarrow \text{Layer}_l(\boldsymbol{H}_{\text{img}}^{(k-1,t)}, \hat{\boldsymbol{H}}_{\text{text}}^{(k-1,t)})$
13:         $\boldsymbol{H}_{\text{text}}^{(k,t)} \leftarrow \hat{\boldsymbol{H}}_{\text{text}}^{(k,t)}[-n :, :]$         ▷ Drop soft tokens
14:     **end for**
15: **end for**
16: **return** $\hat{\mathbf{X}} = \text{Decoder}(\boldsymbol{H}_{\text{img}}^{(N,t_0)})$

---

**Inference Details** A detailed description of the image generation algorithm on MM-Dit is provided in algorithm 1. At each layer, a distinct set of soft tokens is prepended to the text features and used exclusively within that layer. These soft tokens are not carried over to subsequent layers; instead, new soft tokens are introduced or omitted as appropriate. Their primary role is to guide the text tokens toward better text-image alignment, particularly during the early stages of the model.

**Memory Efficiency** For the memory efficiency metrics reported in table 1, values were measured by averaging results over 50 runs with an A100 GPU. Image resolution was set to $512 \times 512$ for SD1.5 and $1024 \times 1024$ for SDXL and SD3.

## C  Prompt for Editing dataset

To generate source/target text prompt from images, we leverage LLaVA [25][1] and Llama [10][2] sequentially. Specifically, we extract source text description from 800 training images of DIV2K dataset by giving each image and following text prompt to LLaVA.

     "Describe the object and background in the image"

---

[1]We use checkpoint from 4bit/llava-v1.5-13b-3GB
[2]We use Llama-3.1-8B-Instruct model.

Then, we generate the target text prompt that has only a single different concept compared with the source text prompt using Llama with the following instruction.

> "You are an AI assistant for generating paired text prompts for real image editing tasks. Your goal is to modify a given text description by replacing an object with other while strictly following these rules:
>
> - 1) Modify only one object (i.e., a single meaningful concept such as an object). It could be small object.
> - 2) The replacement must be significantly different from the original concept but contextually appropriate. Avoid unrealistic substitutions (e.g., changing "rabbit on grass" to "rocket on grass").
> - 3) Ensure diversity in word choices across different modifications.
> - 4) Preserve all other words exactly as they are. Do not change sentence structure, introduce new elements, or modify additional details.
> - 5) Do not provide any additional words—output only the modified text description.
> - 6) Do not change or add colors. Specifically, when modifying a building, change only the appearance, not the type of building (e.g., do not change "building" to "church" or "lighthouse").
> - 7) Modify only one feature at a time. If changing an object (e.g., "starfish" to "sea turtle"), do not alter its color, shape, or other attributes.
>
> Example:
>
> Input: The image features a close-up of a brown dog with a blue nose. The dog is standing in a grassy field, and the background is blurred, creating a focus on the dog's face. The dog's ears are perked up, and its eyes are open, giving it a curious and attentive expression. The dog's fur is brown, and the grass in the background is green, creating a natural and vibrant scene.
>
> Output: The image features a close-up of a brown fox with a blue nose. The fox is standing in a grassy field, and the background is blurred, creating a focus on the fox's face. The fox's ears are perked up, and its eyes are open, giving it a curious and attentive expression. The fox's fur is brown, and the grass in the background is green, creating a natural and vibrant scene."

## D  Discussion on counting metric

**Adopting Soft Tokens Only A Few Layers**   As shown in section 4, our model demonstrates notable improvements in generating single and multiple objects, as well as in color and position-aware synthesis. However, it exhibits a relative decline in accurately generating the specified number of objects described in the text prompts. We hypothesize that this drop in the counting metric stems from the soft tokens excessively emphasizing textual cues, which can lead to overgeneration of object instances. To address this, we limited the use of soft tokens to the early layers of the model. As presented in table 7, applying soft tokens only to layers $1 \sim 2$ preserves strong overall performance while mitigating the degradation in counting accuracy observed when soft tokens are applied across layers $1 \sim 5$.

**Incorporating Counting Loss during training**   To further enhance counting fidelity, we trained SoftREPA with a lightweight object-counting loss. Specifically, we employ a mean squared error (MSE) loss between the predicted object count, which is derived from denoised images, and ground-truth labels using the lightweight object detection module, YOLOv8 [40]. This variant, shown in the last row of table 7, shows improved counting accuracy without sacrificing overall generation quality, suggesting that it effectively reduces the tendency to replicate objects due to textual overemphasis.

## E  Additional Results on T2I Generation

Additional qualitative results for SD1.5 and SDXL on the COCO-val5K dataset are presented in fig. 7 and fig. 8, respectively. Furthermore, qualitative examples on the GenEval benchmark are provided in fig. 9.

| GenEval | | | | | | | | |
|---|---|---|---|---|---|---|---|---|
| Model | Layers | Mean↑ | Single↑ | Two↑ | Counting↑ | Colors↑ | Position↑ | Color Attribution↑ |
| SD3 | | 0.68 | 0.99 | 0.86 | 0.56 | 0.85 | 0.27 | 0.55 |
| SD3 (Ours) | 1 | **0.70** | 0.99 | 0.91 | 0.51 | 0.89 | 0.31 | 0.58 |
| SD3 (Ours) | 1∼2 | 0.69 | **1.00** | 0.89 | 0.50 | 0.88 | **0.34** | 0.56 |
| SD3 (Ours) | 1∼5 | **0.70** | **1.00** | **0.95** | 0.29 | **0.92** | **0.34** | **0.68** |
| SD3 (Ours) + count | 1∼5 | 0.69 | 0.99 | 0.88 | **0.59** | 0.86 | 0.25 | 0.55 |

Table 7: Additional quantitative comparison on the GenEval [9] benchmark. **Bold** indicates the best performance, and underline denotes the second best. "Layers" refers to the transformer layers where soft tokens are applied.

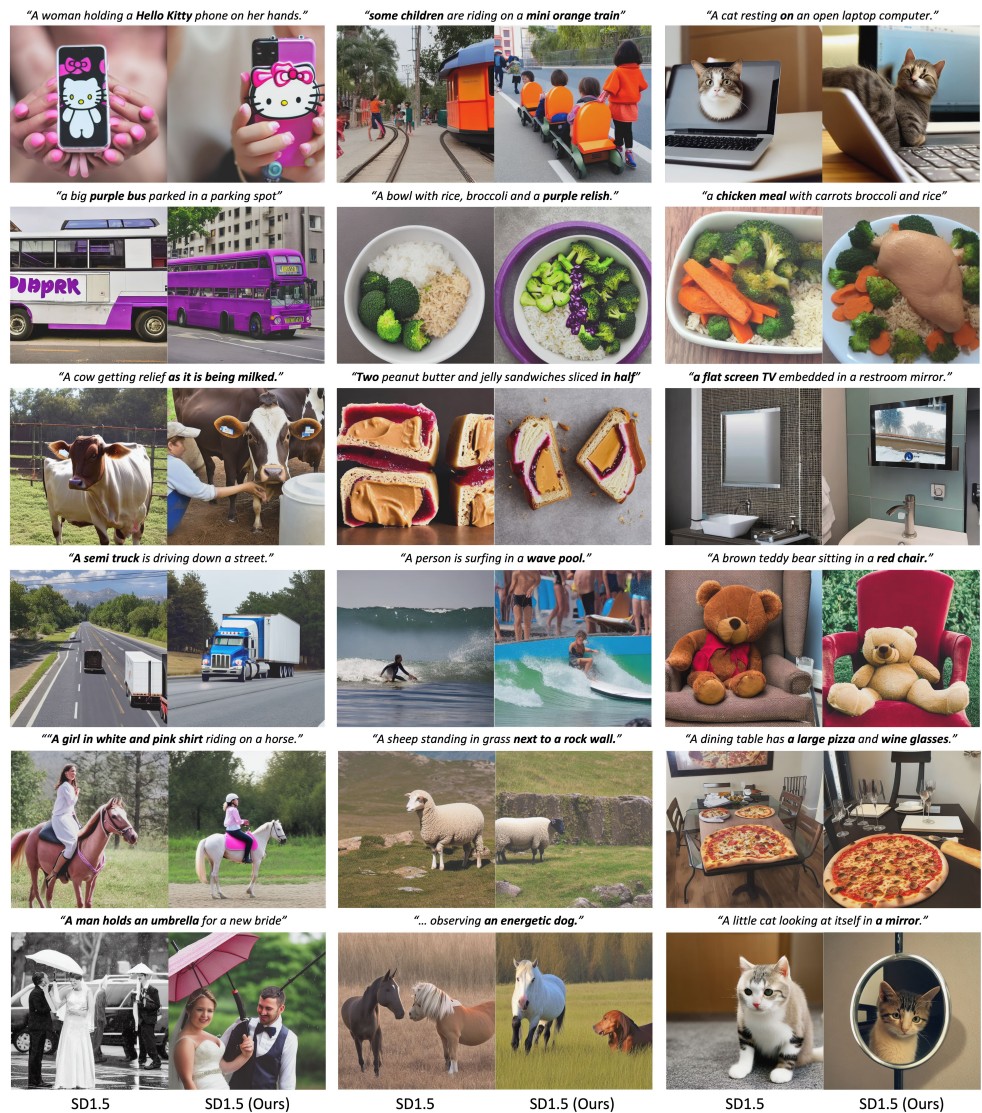

Figure 7: The qualitative results of text-to-image generation comparing SD1.5 and SD1.5 with proposed method. The given text is from COCO dataset.

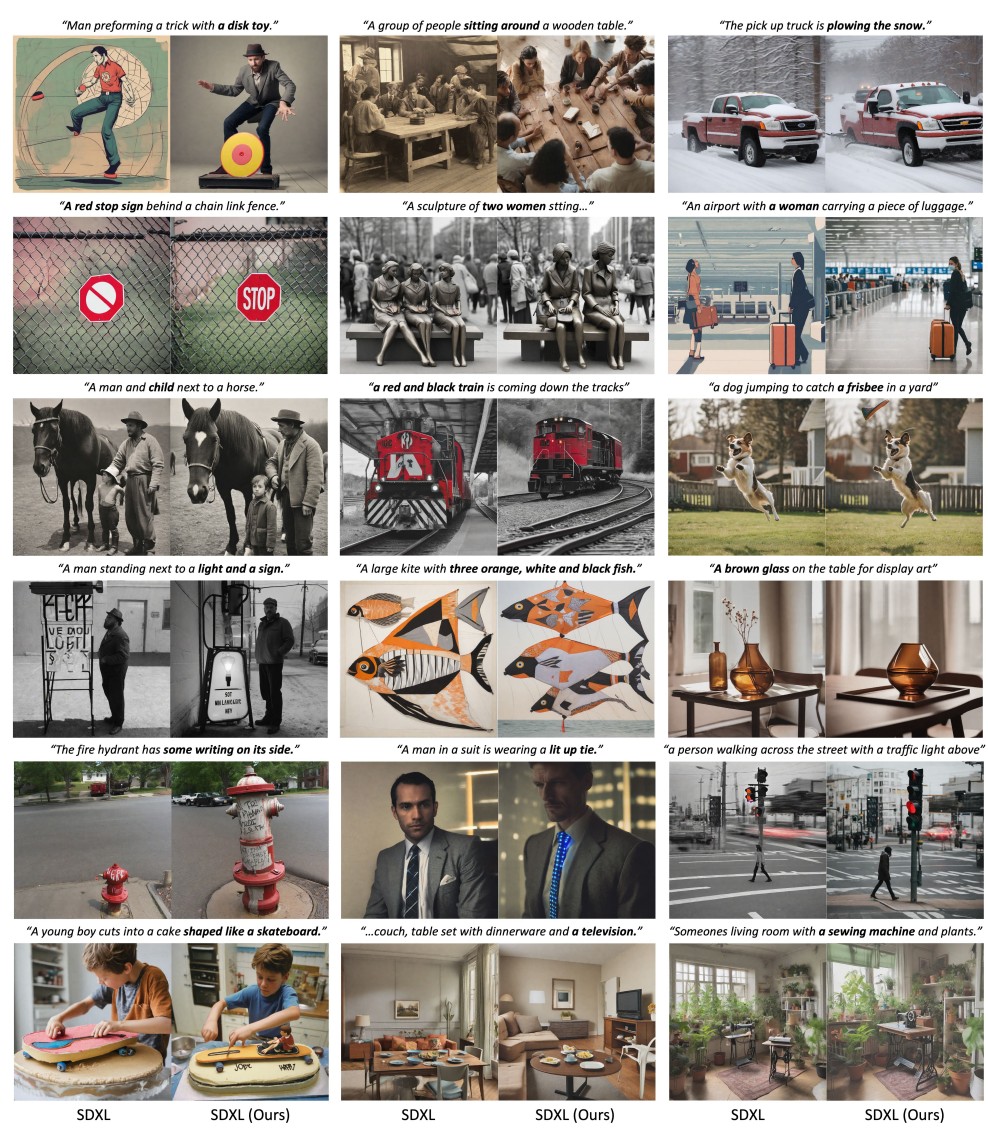

Figure 8: The qualitative results of text-to-image generation comparing SDXL and SDXL with proposed method. The given text is from COCO dataset.

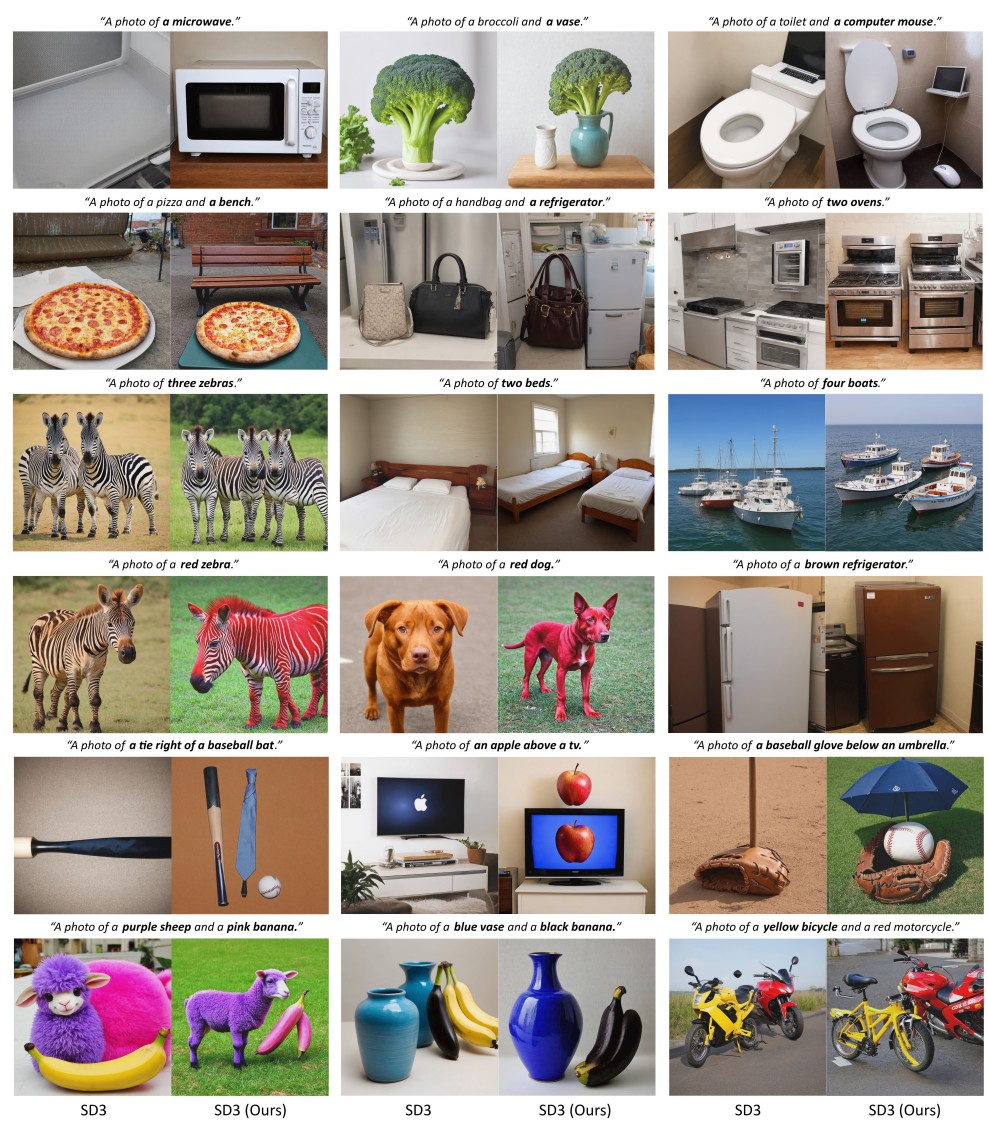

Figure 9: The qualitative results of text-to-image generation comparing SD3 and SD3 with proposed method. The given text is from GenEval dataset.

# F    Additional Results on Image Editing

Qualitative comparisons of SoftREPA across various editing methods, including PnP [38], MasaCtrl [4] with DDIM [35], Direct Inversion [14], and FlowEdit [19], on the PIEBench dataset are presented in fig. 11, corresponding to the quantitative results in table 2.

Quantitative results on the DIV2K and Cat2Dog datasets under varying target CFG scales are reported in table 8 and table 9, respectively. The corresponding qualitative results for different CFG scales are also shown in table 8. Additionally, qualitative examples across different numbers of editing steps are provided in fig. 12 and fig. 10.

| Model | NMAX | target CFG | Human Preference | | Text Alignment | | Image Quality | |
|---|---|---|---|---|---|---|---|---|
| | | | ImageReward↑ | PickScore↑ | CLIP↑ | HPS↑ | FID↓ | LPIPS↓ |
| FlowEdit | 33 | 13.5 | 0.380 | 21.749 | 0.261 | 0.255 | 52.038 | 0.154 |
| FlowEdit | 33 | 16.5 | 0.466 | 21.824 | 0.263 | 0.259 | 55.371 | 0.180 |
| FlowEdit | 33 | 19.5 | 0.510 | 21.869 | 0.265 | 0.261 | 58.069 | 0.200 |
| FlowEdit +Ours | 30 | 9 | 0.466 | 21.884 | 0.263 | 0.260 | 52.633 | 0.149 |
| FlowEdit +Ours | 30 | 11 | 0.564 | 21.985 | 0.268 | 0.265 | 56.900 | 0.178 |
| FlowEdit +Ours | 30 | 13 | 0.627 | 22.050 | 0.272 | 0.270 | 59.710 | 0.201 |

Table 8: Quantitative results of image editing regarding image quality and target text alignment of generated images with various target CFG scales on **DIV2K** dataset.

| Model | NMAX | target CFG | Human Preference | | Text Alignment | | Image Quality | |
|---|---|---|---|---|---|---|---|---|
| | | | ImageReward↑ | PickScore↑ | CLIP↑ | HPS↑ | FID↓ | LPIPS↓ |
| FlowEdit | 33 | 13.5 | 0.937 | 20.726 | 0.225 | 0.266 | 197.1 | 0.199 |
| FlowEdit | 33 | 16.5 | 0.908 | 20.641 | 0.229 | 0.272 | 201.05 | 0.217 |
| FlowEdit | 33 | 19.5 | 0.882 | 20.572 | 0.232 | 0.276 | 202.92 | 0.234 |
| FlowEdit +Ours | 30 | 7 | 1.111 | 21.143 | 0.226 | 0.269 | 198.71 | 0.173 |
| FlowEdit +Ours | 30 | 9 | 1.144 | 21.221 | 0.234 | 0.283 | 208.18 | 0.204 |
| FlowEdit +Ours | 30 | 11 | 1.144 | 21.224 | 0.239 | 0.290 | 214.72 | 0.230 |
| FlowEdit +Ours | 30 | 13 | 1.135 | 21.200 | 0.241 | 0.293 | 219.35 | 0.252 |

Table 9: Quantitative results of image editing regarding image quality and target text alignment of generated images with various target CFG scales on **Cat2Dog** dataset.

*"A close up photo of cat."* → *"A close up photo of dog."*

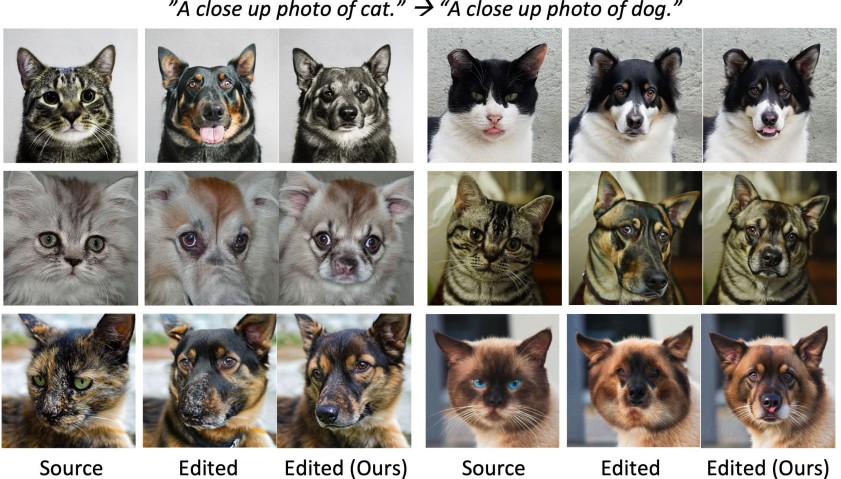

Source          Edited          Edited (Ours)          Source          Edited          Edited (Ours)

Figure 10: Additional qualitative results of text-guided image editing on Cat2Dog dataset.

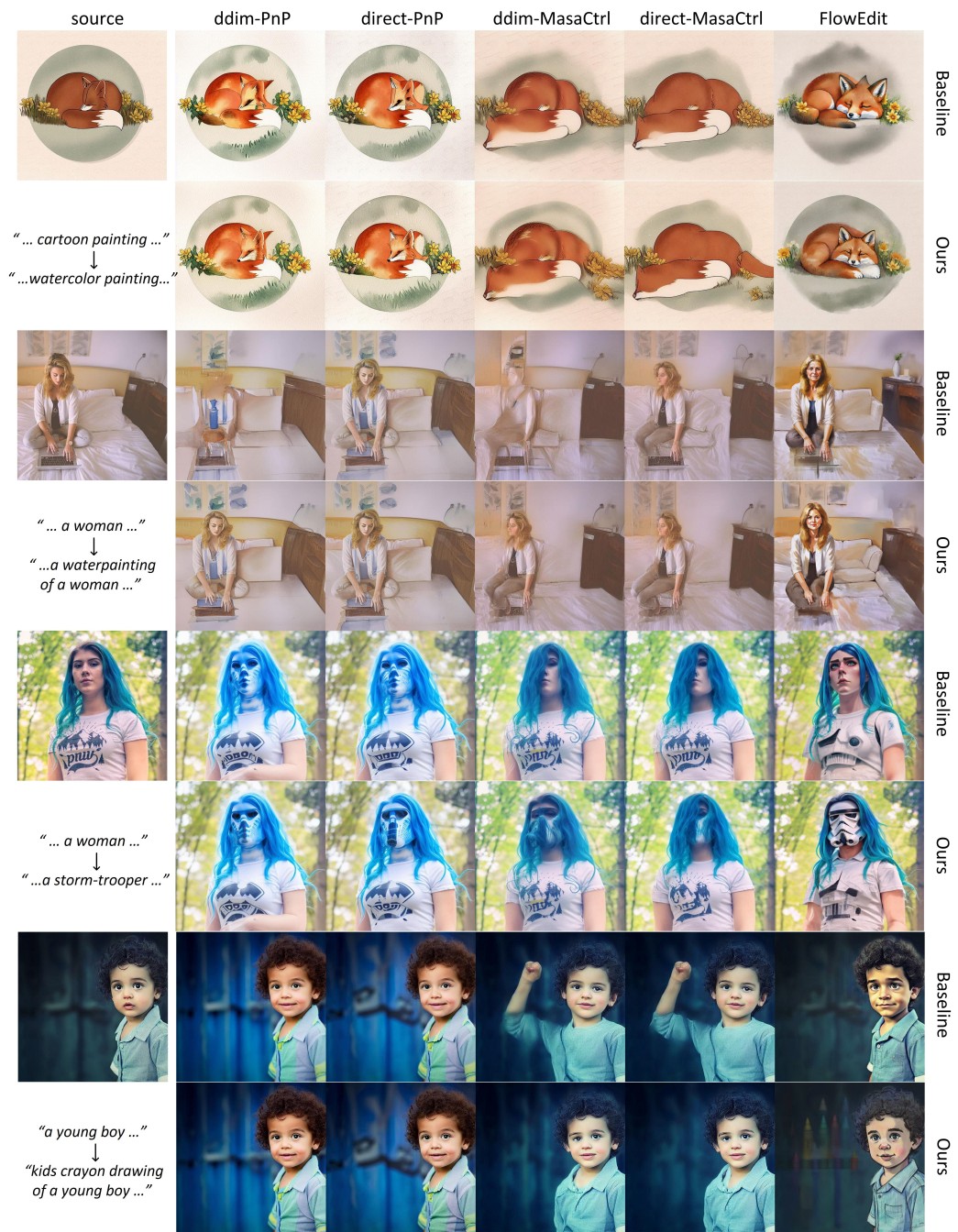

Figure 11: The qualitative results of various editing methods including PnP, MasaCtrl, FlowEdit with DDIM and Direct inversion on PIEBench.

# G    Additional Results on Ablation Study

Quantitative results for varying the number of layers and soft tokens are shown in table 10, with corresponding qualitative examples in fig. 13 and fig. 14. As seen in fig. 13, text prompts are increasingly emphasized as more layers adopt soft tokens. The same input image is used in fig. 14 to illustrate the effect of different token counts.

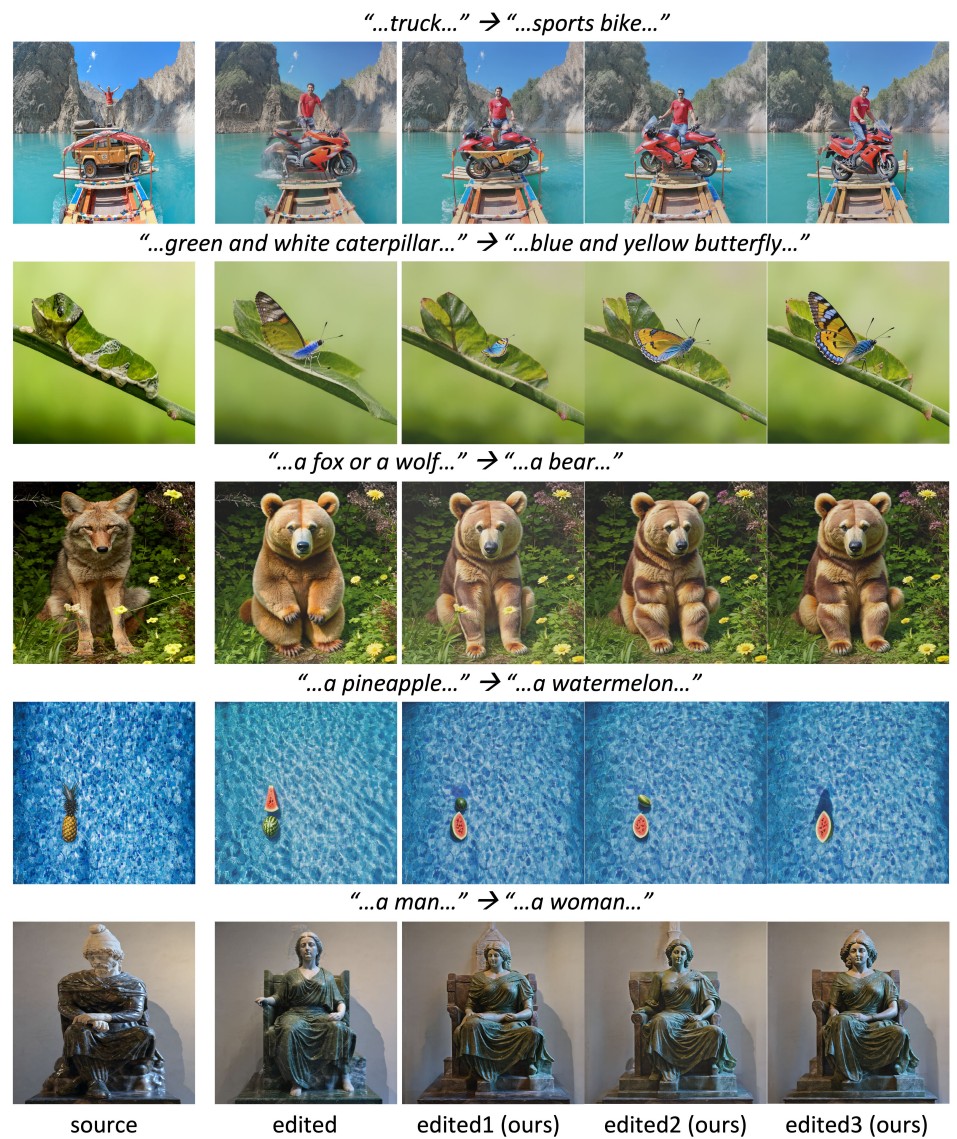

Figure 12: The additional qualitative results on editing using FlowEdit. The edited1-3 using soft tokens are vary on the number of editing steps, $27, 29, 30$ out of $50$ total timesteps, respectively.

## H  Additional Results on comparing with LoRA

In table 11, we conducted various conditions on LoRA finetuning, including top-5-layer and full-layer LoRA, as well as the corresponding number of trainable parameters and the chosen LoRA rank. Specifically, LoRA with top-5 layers uses 0.4M parameters, LoRA applied to all layers uses 2.3M parameters, and our soft token approach uses 0.9M parameters for 5 layers. We adopt a LoRA rank of 4, which is commonly used for diffusion model finetuning.

To enable a direct comparison with a similar number of parameters, we additionally report results for LoRA applied to the top-10 layers (0.9M parameters) and top-5 layers with rank 8 (0.9M parameters). Among these settings, LoRA on the top-5 layers with rank 4 yields the best results within LoRA baselines. Importantly, the combination of Soft Tokens and contrastive Loss consistently achieves the strongest performance across key metrics, supporting our claim that both techniques independently and meaningfully improve text-image alignment.

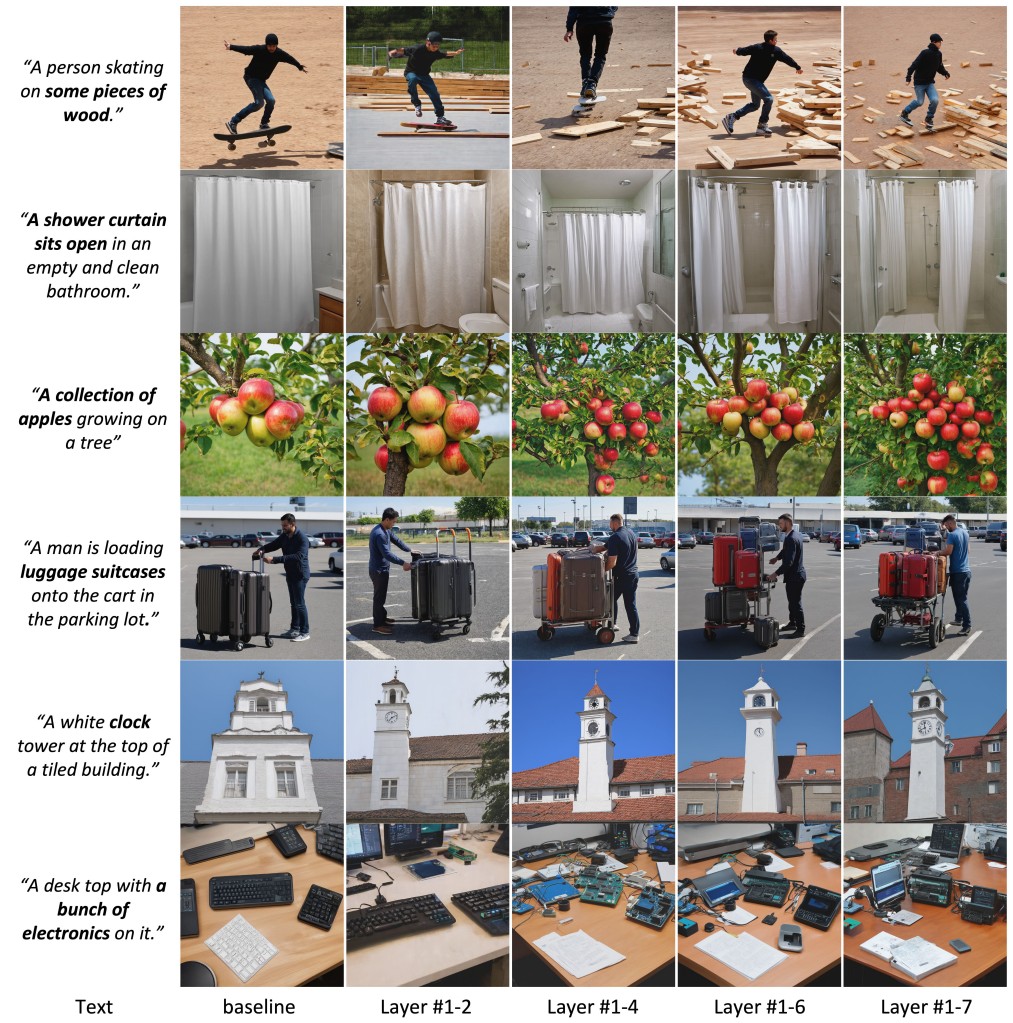

| | Text | baseline | Layer #1-2 | Layer #1-4 | Layer #1-6 | Layer #1-7 |

Figure 13: Qualitative results of the ablation study on the **number of layers** adopting soft tokens.

| # of tokens | # of layers | Human Preference | | Text Alignment | | Image Quality | |
|---|---|---|---|---|---|---|---|
| | | ImageReward↑ | PickScore↑ | CLIP↑ | HPS↑ | FID↓ | LPIPS↓ |
| 8 | 2 | 0.993 | 22.556 | 0.263 | 0.286 | 72.253 | 0.428 |
| 8 | 4 | 1.009 | 22.464 | 0.266 | 0.285 | 72.308 | 0.424 |
| 8 | 6 | 0.984 | 22.403 | 0.267 | 0.286 | 73.840 | 0.427 |
| 8 | 7 | 0.944 | 22.197 | 0.269 | 0.280 | 74.546 | 0.425 |
| 1 | 5 | 1.054 | 22.492 | 0.272 | 0.283 | 58.430 | 0.426 |
| 4 | 5 | 1.063 | 22.493 | 0.269 | 0.287 | 60.766 | 0.429 |
| 8 | 5 | 1.056 | 22.516 | 0.268 | 0.288 | 62.644 | 0.431 |
| 16 | 5 | 0.801 | 22.095 | 0.265 | 0.278 | 60.743 | 0.431 |
| 32 | 5 | 0.675 | 21.876 | 0.257 | 0.275 | 61.623 | 0.426 |

Table 10: Quantitative results of image generation ablation study on the number of soft tokens and the number of layers. The evaluation is conducted on COCO val 1K dataset using SD3 backbone.

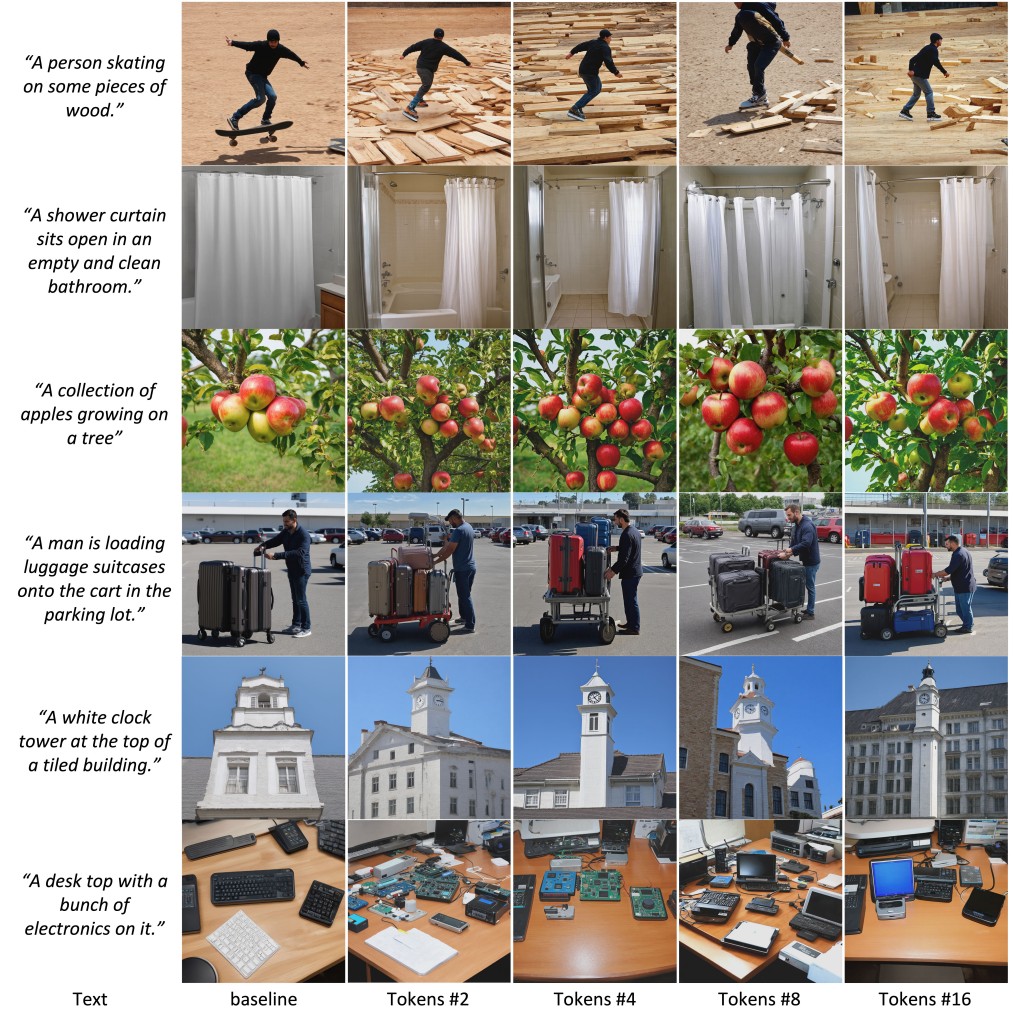

| Text | baseline | Tokens #2 | Tokens #4 | Tokens #8 | Tokens #16 |

Figure 14: Qualitative results of the ablation study on the **number of tokens** adopting soft tokens.

|  | | | | Human Preference | | Text Alignment | | Image Quality | |
| Model | Rank | Layers | # of Params | ImageReward↑ | PickScore↑ | CLIP↑ | HPS↑ | FID↓ | LPIPS↓ |
|---|---|---|---|---|---|---|---|---|---|
| SD3 | | | | 90.89 | 22.50 | 26.26 | 27.97 | **56.87** | **42.47** |
| SD3 + LoRA + Contrastive loss | 4 | 1-5 | 0.4M | 97.72 | **22.57** | 26.42 | 28.44 | 70.33 | 42.71 |
| SD3 + LoRA + Contrastive loss | 8 | 1-5 | 0.9M | 95.92 | 22.55 | 26.28 | 28.28 | 70.30 | 42.55 |
| SD3 + LoRA + Contrastive loss | 4 | 1-10 | 0.9M | 92.72 | 22.54 | 26.23 | 28.10 | 70.53 | 42.59 |
| SD3 + LoRA + Contrastive loss | 4 | all layers | 2.3M | 80.89 | 22.20 | 26.70 | 25.97 | 71.57 | **41.54** |
| SD3 + Soft tokens + Contrastive loss (Ours) | - | 1-5 | 0.9M | **106.32** | 22.49 | **26.92** | **28.70** | 60.76 | 42.99 |

Table 11: Quantitative comparison of T2I generation between soft tokens and various configurations of LoRA on SD3. Generation quality is evaluated on the COCO-val 1K [23]. ImageReward, CLIP, HPS, and LPIPS are scaled by $\times 10^2$.

