# OpenReview forum: "Aligning Text to Image in Diffusion Models is Easier Than You Think"
_NeurIPS.cc/2025/Conference — NeurIPS 2025 poster_

### Official Review · Reviewer_F1wd · 2025-06-09

**Clarity:** 3
**Significance:** 2
**Originality:** 2
**Rating:** 4
**Confidence:** 4

**Summary:**

This paper tries to improve the text-image alignment in diffusion-based T2I model. Following REPresentation Alignment(REPA), the authors introduce contrastive representation loss to diffusion model training. The authors define the similarity measure using the the variation of the denoising score matching loss and introduce extra soft tokens at each layers. The experimental results show improvement of the finetuned model over the baseline model.

**Questions:**

1. According to equation 10 and 11, the similarity measure is defined as the variance of denoising score matching loss. As an expectation calculation is included in the equation, do the authors estimate the expectation with Monte Carlo method? If so, how many samples are needed for this estimation for one text-image pair? And will the number of samples (using in Monte Carlo estimation) influenced the performance? When doing the estimation, what is the distribution for timestep sampling?

2. What is the "time scheduling parameter" $\tau(t)$ used in the paper? Are the experimental results sensitive to the choice of $\tau(t)$?

3. What data source do the author use to train the model?

**Ethical Concerns:**

["NO or VERY MINOR ethics concerns only"]

**Final Justification:**

The rebuttal has addressed my concerns. I would like to increase my rating to 4.

**Limitations:**

Yes, the authors have adequately addressed the limitations and potential negative societal impact of their work.

**Paper Formatting Concerns:**

From my perspective, there is no major formatting issues in this paper. But it seems that many important training details (choice of $\tau (t)$, training data, sampling strategy) are not included in the main paper.

**Quality:**

2

**Strengths And Weaknesses:**

***Strengths***

The idea of introducing contrastive representation loss makes senses. And the experimental results suggest improvement over the baseline model.


***Weaknesses***

a. The main idea of introducing contrastive representation directly follow REPresentation Alignment(REPA) [1]. And implementation-wise,  introducing and optimizing some special text token are also not new in diffusion model (e.g. [2][3])

b. The design of the similarity measure is new for this paper. However, I have some concerns regarding this design choice:

1. The original REPA [1] work introduces the pretrained self-supervised visual representations to calculate the similarity. Yet this work purely relies on the diffusion model itself to define the similarity. Using pretrained self-supervised visual representations make more sense to me as the pretrained model brings extra knowledge during the training. It might be better if the authors can experimentally compare their design with the one using pretrained self-supervised visual representations (i.e. original REPA).

2. Also, the authors mainly compare their finetuned model with the unfinetuned baseline model. I think it is worth adding an ablation which keeps the structure design of the soft-token, but finetune the model with the original diffusion loss and same data. This mainly eliminates the possibility that the improvement comes from continue training on high quality data and can demonstrate that contrastive learning really plays an important role here.

3. This design introducing several new hyper-parameters, including the number of soft tokens, the time scheduling parameter, the layer to include the soft tokens. Also, in appendix B, it seems that for different models, one might need to decide whether to include original denoising loss. These make me concern about the generalizability of the proposed design.

[1] Representation Alignment for Generation: Training Diffusion Transformers Is Easier Than You Think
[2] Null-text Inversion for Editing Real Images using Guided Diffusion Models
[3] DreamBooth: Fine Tuning Text-to-Image Diffusion Models for Subject-Driven Generation

---

> ### Author Rebuttal · Authors · 2025-07-31
>
> ### **Wa: Novelty over REPA and Prompt Optimization**
> We thank the reviewer for the opportunity to clarify our contributions. While inspired by REPA [1], our method differs in both objective and application. REPA aims to accelerate training by aligning DiT features with external discriminative representations.  In contrast, our work targets text-image alignment by implicitly aligning internal representations of two different domains through contrastive learning.
>
> There are several works from prompt optimization. CoOP [2] introduces learnable soft prompts for CLIP models at the input level by replacing fixed prompt templates (e.g., “a photo of a [CLS]”) with learnable embeddings that help differentiate between predefined class labels. However, our approach generalizes this idea by adopting soft tokens within the model's internal layers, independent of predefined classes. This extension enables fine-grained, layer-wise modulation of semantic alignment signals.
>
> To our knowledge, this is the **first attempt to optimize soft tokens in diffusion models by modulating internal features beyond input embeddings**. Our results show that these tokens significantly improve generative quality conditioning on given text by influencing feature propagation.
>
> To validate this design, we compare with LoRA applied to the top 5 layers, which outperforms full-layer tuning. As shown in **Table 1**, our method—*Soft Tokens + Contrastive Loss*—achieves the best performance on *ImageReward*, *CLIP*, *HPS*, and *FID*, while remaining competitive on *PickScore* and *LPIPS*, demonstrating the complementary strength of our architectural and training choices.
>
> **Table 1. Quantitative comparison of parameter-efficient tuning methods**
> | Method                        | ImageReward | PickScore |  CLIP  |  HPS  |  FID  |  LPIPS  |
> |-------------------------------|--------------------|----------------|----------|---------|---------|-----------|
> | SD3                              | 90.89               | 22.50         | 26.26  | 27.97 | **56.87** | **42.47** |
> | LoRA + Contrastive loss | 97.72             | **22.57**    | 26.42  | 28.44 | 70.33 | 42.71 |
> | Soft token + Contrastive loss (Ours) | **106.32** |22.49 | **26.92**  | **28.70** | 60.76 | 42.99 |
>
> **References**
> [1] Yu, Sihyun, et al. "Representation alignment for generation: Training diffusion transformers is easier than you think." arXiv preprint arXiv:2410.06940 (2024).
>   [2] Kaiyang Zhou, Jingkang Yang, Chen Change Loy, and Ziwei Liu. Learning to prompt for vision-language models. International Journal of Computer Vision, 130(9):2337–2348, 2022.
>
> ---
> ### **Wb1: Compare the design with the original REPA**
> We sincerely thank the reviewer for the thoughtful suggestion. We agree that pretrained self-supervised visual encoders provide powerful external knowledge, and we appreciate the insight into comparing with such representations as explored in REPA [1]. However, we would like to highlight a key distinction in the **training objectives** and **representation targets** between REPA and our proposed method.
>
> REPA aims to accelerate **pretraining stage** by explicitly aligning the internal features of the diffusion model (DiT) with external discriminative features obtained from a pretrained visual encoder. This is feasible because the internal generative features and the external discriminative features exhibit natural similarity in the image domain.
>
> In contrast, SoftREPA is designed to enhance text-image alignment of **pretrained models** by introducing learnable soft text tokens that explore a richer textual feature space within the model. As the reviewer pointed out, our method operates entirely within the latent space of the diffusion model itself, relying solely on the model’s internal priors. Since the soft tokens are applied to textual features, there is no inherently sufficient alignment with visual representations from external encoders, making direct supervision from such representations less applicable in our setting.
>
> That said, we believe the reviewer’s suggestion opens up an interesting future direction. It may be promising to explore soft image token tuning using pretrained visual features, which could potentially enhance the image representation space in a complementary way. While this is beyond the scope of our current work, we are excited by this possibility. Once again, we thank the reviewer for the thoughtful suggestion.
>
> ---
> ### **Wb2: Finetuning Baseline Without Contrastive Loss**
> We thank the reviewer for raising this insightful point, which directly addresses the contribution of soft tokens and the contrastive learning objective. To further clarify their individual roles, we conducted an additional ablation study as suggested.
>
> To assess whether the observed improvements stem from continued training on high-quality data rather than the contrastive loss itself, we performed a new ablation where the model retains the soft token architecture but is fine-tuned using the original denoising score matching (DSM) loss on the same dataset. This setup isolates the effect of contrastive learning from soft tokens. As shown in **Table 2**, this variant consistently underperforms compared to our full method, confirming that the gains are not merely due to extended training.
>
> **Table 2. Quantitative comparison of extra DSM training**
> | Method                        | ImageReward | PickScore |  CLIP  |  HPS  |  FID  |  LPIPS  |
> |-------------------------------|--------------------|----------------|----------|---------|---------|-----------|
> | SD3                              | 90.89               | **22.50**         | 26.26  | 27.97 | **56.87** | **42.47** |
> | Soft token + DSM loss  | 86.98               | 22.39        | 26.29  | 27.78 | 70.39 | 42.65 |
> | Ours | **106.32** |22.49 | **26.92**  | **28.70** | 60.76 | 42.99 |
>
> ---
> ### **Wb3: Concerns About Hyperparameters and Generalization**
>
> We would like to clarify that although new parameters are introduced, we observed **consistent trends** across different models (SD3, SD1.5, SDXL). For instance:
> - In terms of layer placement, all models showed best performance when soft tokens were applied only to earlier blocks (SD3) or down blocks of UNet(SD1.5, SDXL).
> - The time scheduling parameter was kept constant across all models, and we did not observe high sensitivity to its value.
> - The optimal number of soft tokens for SD3 lies between 4 and 8, and this range also covers the optimal settings for SD1.5 and SDXL, which show similarly strong performance across a broader range.
>
> These consistent tendencies suggest that our method behaves robustly across architectures, requiring minimal model-specific tuning—similar to other generation techniques like classifier-free guidance (CFG).
>
> Furthermore, to evaluate the generalizability of SoftREPA, we conducted experiments on SD1.5 and SDXL by training only with the contrastive loss, **omitting the denoising score matching (DSM)** term entirely. As shown in **Table 3**, even without DSM, the model performs competitively, further supporting that our contrastive learning framework is effective and broadly applicable.
>
> **Table 3. Quantitative comparison on the generalizability of SoftREPA**
> | model | loss term  | ImageReward | PickScore |  CLIP  |  HPS  |  FID  |  LPIPS  |
> |----------|---------------|--------------------|----------------|----------|---------|---------|-----------|
> | SD1.5    | baseline                  | 17.72 | 21.47   | 26.45  | 25.08 | 24.59 | 43.80 |
> | SD1.5    | DSM + Contrastive | **32.89** | 21.50  | **27.33**  | 25.18 | **23.43** | **43.38** |
> | SD1.5    | Contrastive             | 32.63 | **21.61**    | 27.10  | **25.94** | 29.25 | 44.09 |
>
> ---
> | model | loss term  | ImageReward | PickScore |  CLIP  |  HPS  |  FID  |  LPIPS  |
> |----------|---------------|--------------------|----------------|----------|---------|---------|-----------|
> | SDXL    | baseline                  | 75.06 | 22.38    | 26.76  | 27.35 | **24.69** | **42.05** |
> | SDXL    | DSM + Contrastive | **85.29** | **22.62**    | 26.80  | 28.30 | 26.04 | 42.39 |
> | SDXL    | Contrastive             | 81.09 | 22.59    | **26.87**  | **28.32** | 26.42 | 42.69 |
>
> ---
> ### **Q1: Expectation Estimation via Monte Carlo**
> As described in Section 3.2 of the paper, we approximate the expectation in the logit computation l(x,y) using a single Monte Carlo sample for efficiency. To further stabilize training, we use the same sampled timestep and noise across all pairs within a minibatch when computing the loss.
> During training, the timestep ttt is sampled from a uniform discrete distribution over [0,999], which is a standard choice in diffusion-based models. During inference, we use a fixed sequence of timesteps determined by the scheduler: DPM-Solver++ for SD3, and DDIM for SD1.5 and SDXL.
> We did not observe significant performance degradation with a single sample in practice, though increasing the number of samples could potentially improve stability at the cost of computational overhead.
>
> ---
> ### **Q2: Time Scheduling Parameter $\tau(t)$**
> The scheduling parameter $\tau(t)$ used in constructing the SoftREPA logit is set to a constant value of 0.07, which serves as a temperature parameter to scale the logits. This value is commonly used in contrastive learning literature and was found to work well empirically. We did not observe strong sensitivity to this hyperparameter in our experiments.
>
> ---
> ### **Q3: Data Source for Training**
>  We used the COCO dataset to train the soft tokens, as described in Section 4 (Text-to-Image Generation) of the paper.

---

> > ### Comment · Reviewer_F1wd · 2025-08-03
> >
> > Thank the authors for the reply. My concerns have resolved and i would like to increase my score to 4.

---

> > > ### Author Response · Authors · 2025-08-03
> > > **Thank you for increasing your score!**
> > >
> > > Thank you very much for increasing your score. We’re pleased to hear that your original concerns have been properly addressed.

---

### Official Review · Reviewer_v3Q6 · 2025-07-01

**Clarity:** 4
**Significance:** 3
**Originality:** 3
**Rating:** 4
**Confidence:** 3

**Summary:**

The paper introduces SoftREPA; a novel and lightweight fine-tuning method for improving t2i alignment in pre-trained t2i diffusion models. The method freezes the pre-trained models and introduces a small number of learnable "soft tokens". These learnable tokens are optimized with contrastive loss that leverages the denoising objective to better differentiate between positive and negative text-image pairs. The authors provide theoretical analysis by showing that this approach is related to maximizing the mutual information between the text and image pairs. The authors demonstrated empirically an enhanced semantic consistency for both image generation and editing tasks.

**Questions:**

As discussed in the weaknesses above.

1. How would you choose a finetuning dataset especially that Table 1. shows significant drop in counting.
2. How would you better compare this method to other relevant methods? For instance, the author mentioned several training-free techniques (like CFG++) which could be potential candidates to compare to.

**Ethical Concerns:**

["NO or VERY MINOR ethics concerns only"]

**Limitations:**

Yes

**Paper Formatting Concerns:**

No concern

**Quality:**

4

**Strengths And Weaknesses:**

Strengths

* Pluggable light-weight module to pretrained t2i models which can used to improve t2i alignment with light-weight finetuning.
* Repurposing the flow matching network output for contrastive learning.
* Provide convincing theoretical analysis that SoftREPA is implicitly maximizes the mutual information between text-image pairs.

Weaknesses

* The paper didn't discuss how the choice of positive/negative pairs will impact the learning, not how to select a finetuning dataset.
* The paper didn't provide convincing comparisons to relavant techniques, instead they compared to RankDPO which is orthogonal method.
* Table 1. shows significant drop in counting metric.

---

> ### Author Rebuttal · Authors · 2025-07-31
>
> ---
> ### **W1W3Q1: Lack of Discussion on Positive/Negative Pair Selection, Counting Drop, and Dataset Choice**
> Per reviewer's request, we conducted an additional experiment by training SoftREPA with hard negative samples. For each COCO caption, we automatically generated three hard negatives using GPT4.1-mini, by minimally modifying the original caption with respect to: **Counting** (e.g., altering the number of objects), **Color** (e.g., changing the color of one object), **Position** (e.g., modifying spatial arrangement). During training, for each training sample, these hard negatives were incorporated alongside other negative samples used in our original method.
>
> As shown in ***Table 1***, we did not observe a significant improvement from adding hard negatives to the training data. This may be due to the limited number of hard negatives, as we generated only one altered caption per attribute. We hope to conduct further experiments on the impact of hard negatives in future work.
>
> **Table 1: Effect of Incorporating Hard Negatives into Our Method**
> | Method                        | Mean | Single | Two | Counting | Colors | Position | Color Attr.|
> |-------------------------------|----------|--------|----------|-----------|-----------|------------|---------------|
> | SD3                              | 0.68  |  0.99  | 0.86    | **0.56**     | 0.85     |   0.27     |    0.55      |
> | Ours                            | **0.70**  |  **1.00**   | **0.95**    | 0.29     | **0.92**     |   **0.34**     |   **0.68**      |
> | Ours+Hardneg          | 0.68  |  **1.00**   | 0.86    | **0.56**     | 0.84     |   0.28     |   0.53      |
>
>
> ---
> ### **W2Q2: Comparisons to Relevant Methods**
> To provide a more comprehensive comparison, we conducted additional experiments covering both **(1) parameter-efficient fine-tuning methods** and **(2) approaches aimed at improving text-to-image alignment**.
>
> **(1)** In terms of lightweight tuning strategies, we compare *Soft Tokens* with *LoRA*, applied to the top 5 transformer layers for optimal performance. As shown in **Table 2**, the combination of *Soft Tokens + Contrastive Loss* consistently outperforms LoRA-based tuning across multiple metrics, including *ImageReward*, *CLIP*, *HPS*, and *FID*, while performing competitively on *PickScore* and *LPIPS*. This confirms the effectiveness of our architectural design in conjunction with contrastive learning.
>
> **(2)** Regarding alignment-oriented approaches, we would like to clarify that contrastive learning is strongly relevant to Reinforcement Learning from Human Feedback(RLHF) and Direct Preference Optimization (DPO) frameworks [1]. Recent work [1] shows that both RLHF and DPO can be reframed as contrastive learning objectives via mutual information maximization. Since our approach shares the underlying principle, we compare SoftREPA against RankDPO [2] and additionally CaPO [3] in **Table 3**. CaPO [3] is a recently proposed diffusion-RL method built upon SD3, which is reported to outperform Diffusion-DPO [4], as noted in our related works.
>
> Additionally, we include CFG++ [5], **a training-free technique** that improves generation quality by refining the classifier-free guidance step. Since CFG++ is originally implemented on SD1.5 and SDXL, we followed the official implementation details provided in their repository.
>
> We evaluate all models on the GenEval benchmark. As presented in **Table 3**, our method outperforms baselines on several key categories, including *Single*, *Two*, *Colors*, *Position*, and *Color Attribute*, confirming that SoftREPA not only provides an effective tuning method but also achieves strong performance in challenging alignment settings.
>
> ---
> **Table 2. Quantitative comparison of parameter-efficient tuning methods**
>
> | Method                        | ImageReward | PickScore |  CLIP  |  HPS  |  FID  |  LPIPS  |
> |-------------------------------|--------------------|----------------|----------|---------|---------|-----------|
> | SD3                              | 90.89               | 22.50         | 26.26  | 27.97 | **56.87** | **42.47** |
> | LoRA + Contrastive loss | 97.72             | **22.57**    | 26.42  | 28.44 | 70.33 | 42.71 |
> | Soft token + Contrastive loss | **106.32** |22.49 | **26.92**  | **28.70** | 60.76 | 42.99 |
> ---
> **Table 3. GenEval benchmark comparing related works**
> | Method                        | Mean | Single | Two | Counting | Colors | Position | Color Attr.|
> |-------------------------------|----------|--------|----------|-----------|-----------|------------|---------------|
> | SD3                              | 0.68  |  0.99  | 0.86    | 0.56     | 0.85     |   0.27     |    0.55      |
> | CFG++[5]  (w=0.6)      | 0.64  |  0.98   | 0.79    | 0.51     | 0.83     |   0.21     |   0.52      |
> | CFG++[5]  (w=0.8)       | 0.67  |  0.98   | 0.83    | 0.53     | 0.84     |   0.26     |   0.58      |
> | RankDPO[2]                 | 0.74  |**1.00**  | 0.90    | **0.72** | 0.87     |   0.31     |   0.66  |
> | CaPO[3]                      | 0.71  |  0.99  | 0.87    | 0.63     | 0.86     |   0.31     |    0.59      |
> | Ours                            | 0.70  | **1.00** | **0.95** | 0.29  | **0.92**| **0.34** | **0.68** |
>
> ---
> **References**
> [1] Lv, Xufei, et al. "The Hidden Link Between RLHF and Contrastive Learning." arXiv preprint arXiv:2506.22578 (2025).
> [2] Karthik, Shyamgopal, et al. "Scalable ranked preference optimization for text-to-image generation." arXiv preprint arXiv:2410.18013 (2024).
> [3] Lee, Kyungmin, et al. "Calibrated multi-preference optimization for aligning diffusion models." Proceedings of the Computer Vision and Pattern Recognition Conference. 2025.
> [4]Wallace, Bram, et al. "Diffusion model alignment using direct preference optimization." Proceedings of the IEEE/CVF Conference on Computer Vision and Pattern Recognition. 2024.
> [5] Chung, Hyungjin, et al. "Cfg++: Manifold-constrained classifier free guidance for diffusion models." arXiv preprint arXiv:2406.08070 (2024).

---

> > ### Comment · Reviewer_v3Q6 · 2025-08-04
> > **Post-rebuttal review**
> >
> > I'd like to thank the authors for their rebuttal and addressing my concerns. I still think a more comprehensive comparisons are needed and explanation of the drop in counting. For that, I will stay with my initial scores.

---

### Official Review · Reviewer_hkeq · 2025-07-03

**Clarity:** 3
**Significance:** 3
**Originality:** 3
**Rating:** 4
**Confidence:** 3

**Summary:**

In this paper, the authors approach the task of improving text image alignment in generative models. The paper proposes a method termed SoftREPA that attempts to align the features using a contrastive objective.
More concretely, the authors introduce “soft text tokens” that are concatenated to the regular text tokens, and fine tuned using a contrastive objective, where the positive pair corresponds to the aligned text image pair, and the negative samples correspond to unaligned text-image pairs. The soft text tokens vary at different layers and different denoising steps, and are the only learnable parameter, the rest of the model is kept frozen.
The authors also show that minimizing this contrastive objective resembles maximizing the mutual information between the text and image pairs.

**Questions:**

My questions for the authors mostly stem from the weaknesses i listed above.
- could the authors comment on the need for / importance of the soft text tokens.
- How is the LPIPS metric implemented for measuring the quality of text-to-image generation?

**Ethical Concerns:**

["NO or VERY MINOR ethics concerns only"]

**Final Justification:**

My initial concerns about the importance of the soft text tokens was addressed and therefore i keep my original score of borderline accept.

**Limitations:**

yes

**Paper Formatting Concerns:**

I did not find any major formatting issues with the paper.

**Quality:**

2

**Strengths And Weaknesses:**

Strengths:
- The contrastive loss to align the features is well motivated and the idea of using learnable soft text tokens is interesting and unique, that I have not seen in prior papers.
- The method proposed in this paper is quite general, and the authors demonstrate this by applying it to many different base model architectures (SD1.5, SDXL, SD3), and two tasks (text-to-image and image editing).

Weaknesses:
- Importance of the soft text tokens. It is unclear to me whether the improvement shown in the paper is primarily due to the contrastive objective or because of the use of soft text tokens. It is unclear if you would get similar, better, or worse results if you simply finetune the generative models with a small LoRA.

- Evaluation of text-to-image generation. I have a couple of doubts based on the quantitative results shown in table 1. First is that the authors only compare to the pre-trained non-finetuned model for evaluating the human preference, text alignment, and image quality. Even in this limited comparison, it seems like the base model has higher image quality.
Second, how is LPIPS used to evaluate the image quality? LPIPS is a network that is used to compare two images. It is unclear to me how this can be used to evaluate the quality of a text-to-image model. Perhaps the authors can add some implementation details about this metric.

- The authors discuss several other related works that attempt this task of improving the text-image alignment (cfg++, attend-and-excite, DOODL, DiffusionDPO, …). However the authors do not compare to any of these prior works in Table 1. It would be useful to see how the proposed method compares to other methods in the literature.

---

> ### Author Rebuttal · Authors · 2025-07-31
>
> ### **W1/Q1: Necessity and Role of Soft Text Tokens**
>
> We appreciate the reviewer’s question regarding the specific contribution of soft text tokens apart from the contrastive learning objective. To investigate this, we performed an ablation study using **LoRA** as an alternative parameter-efficient tuning method in place of soft tokens. We applied LoRA to the top 5 transformer layers, as this configuration proved more effective than applying LoRA across all layers.
>
> As shown in **Table 1**, the combination of *Soft Tokens + Contrastive Loss* consistently achieves stronger results across multiple metrics, including *ImageReward*, *CLIP*, *HPS*, and *FID*, and performs comparably on *PickScore* and *LPIPS*. This demonstrates that both the architectural design of **soft tokens and the contrastive loss independently contribute to improved text-image alignment**.
>
> In addition, we would like to clarify the role and **motivation behind the soft text tokens**. Our architectural design is primarily inspired by CoOP [1], which proposed a prompt learning strategy for CLIP-based vision-language models that replaces the fixed prompt (“a photo of a [CLS]”) with learned soft prompts capable of distinguishing among a set of predefined classes. This suggests that prompt-tuning enables the model to explore richer semantic embeddings. While CoOP focuses on learning prompts in the input space, we generalize this idea by propagating learned soft tokens through the network, allowing the model to align text and image representations in the internal feature space without relying on predefined class labels.
>
> This architectural choice enables a more flexible and implicit alignment mechanism, encouraging the model to maximize the mutual information between text and image embeddings by refining the internal feature dynamics of the model.
>
> **Table 1. Quantitative comparison of parameter-efficient tuning methods**
> | Method                        | ImageReward | PickScore |  CLIP  |  HPS  |  FID  |  LPIPS  |
> |-------------------------------|--------------------|----------------|----------|---------|---------|-----------|
> | SD3                              | 90.89               | 22.50         | 26.26  | 27.97 | **56.87** | **42.47** |
> | LoRA + Contrastive loss | 97.72             | **22.57**    | 26.42  | 28.44 | 70.33 | 42.71 |
> | Soft token + Contrastive loss | **106.32** |22.49 | **26.92**  | **28.70** | 60.76 | 42.99 |
>
> **References**
> [1] Kaiyang Zhou, Jingkang Yang, Chen Change Loy, and Ziwei Liu. Learning to prompt for vision-language models. International Journal of Computer Vision, 130(9):2337–2348, 2022.
>
> ---
> ### **W2Q2: Clarification on LPIPS and Image Quality**
> We appreciate the reviewer’s concern regarding the use of LPIPS and other image quality metrics. In our study, we computed the LPIPS metric to measure the perceptual distance between the generated image and the real image corresponding to the given prompt. While LPIPS can indicate the similarity between the target and source images in editing tasks, we initially believed it could also reflect image quality in generation tasks. However,  other metrics such as ImageReward, PickScore, and HPS, which effectively reflect image quality,  demonstrate that our method preserves image quality while improving text-image alignment.
>
> ---
> ### **W3: Lack of Comparison with Alignment-Improving Baselines**
>
> We would like to note that our approach is grounded in contrastive learning, which shares a deep theoretical connection with recent preference optimization methods in diffusion models. As highlighted in recent work [1], both Reinforcement Learning from Human Feedback (RLHF) and Direct Preference Optimization (DPO) can be reinterpreted as forms of mutual information maximization—a foundational principle also underlying contrastive learning. Motivated by this connection, we compare our method, SoftREPA, with the diffusion-RL-based methods RankDPO [2] and CaPO [3], which are built on SD3 backbone. Notably, CaPO has been reported to outperform Diffusion-DPO [4], which we cited in our related work.
>
> In addition, we include CFG++ [5], a training-free approach that improves generation quality by refining the classifier-free guidance (CFG) step. Although CFG++ is implemented on SD1.5 and SDXL, we follow the official guidance provided in their public repository to ensure a consistent comparison.
>
> To evaluate alignment quality, we report results on the GenEval benchmark. As shown in **Table 2**, our method performs competitively overall and demonstrates superior results in several categories, including *Single*, *Two*, *Colors*, *Position*, and *Color Attribute*. These results support that SoftREPA offers a robust and effective framework for improving text-image alignment compared to both preference-optimized and training-free baselines.
>
>
> **Table 2. GenEval benchmark comparing related works**
> | Method              | Mean | Single | Two  | Counting | Colors | Position | Color Attr. |
> |---------------------|-------|--------|------|----------|--------|----------|--------------|
> | SD3                 | 0.68  | 0.99   | 0.86 | 0.56     | 0.85   | 0.27     | 0.55         |
> | CFG++ [5] (w=0.6)   | 0.64  | 0.98   | 0.79 | 0.51     | 0.83   | 0.21     | 0.52         |
> | CFG++ [5] (w=0.8)   | 0.67  | 0.98   | 0.83 | 0.53     | 0.84   | 0.26     | 0.58         |
> | RankDPO [2]         | 0.74  | **1.00** | 0.90 | **0.72** | 0.87   | 0.31     | 0.66         |
> | CaPO [3]            | 0.71  | 0.99   | 0.87 | 0.63     | 0.86   | 0.31     | 0.59         |
> | **Ours**            | 0.70  | **1.00** | **0.95** | 0.29     | **0.92** | **0.34** | **0.68**     |
>
> ---
>
> **References**
> [1] Lv, Xufei, et al. *The Hidden Link Between RLHF and Contrastive Learning.* arXiv preprint arXiv:2506.22578 (2025).
> [2] Karthik, Shyamgopal, et al. *Scalable Ranked Preference Optimization for Text-to-Image Generation.* arXiv preprint arXiv:2410.18013 (2024).
> [3] Lee, Kyungmin, et al. *Calibrated Multi-Preference Optimization for Aligning Diffusion Models.* CVPR 2025.
> [4] Wallace, Bram, et al. *Diffusion Model Alignment Using Direct Preference Optimization.* CVPR 2024.
> [5] Chung, Hyungjin, et al. *CFG++: Manifold-Constrained Classifier-Free Guidance for Diffusion Models.* arXiv preprint arXiv:2406.08070 (2024).

---

> > ### Comment · Reviewer_hkeq · 2025-08-08
> >
> > I thank the authors for the responses. After reading the other discussions and the rebuttal, I am inclined towards keeping my score of borderline acceptance.

---

### Official Review · Reviewer_2H9G · 2025-07-03

**Clarity:** 3
**Significance:** 2
**Originality:** 2
**Rating:** 4
**Confidence:** 4

**Summary:**

This work takes a different approach by focusing on representation alignment, inspired by the success of REPA. The authors propose SoftREPA, a lightweight contrastive fine-tuning method that uses soft text tokens to improve alignment with minimal computational cost (less than 1M additional parameters). Theoretical analysis shows it enhances mutual information between text and image features, leading to better semantic consistency. Experiments confirm its effectiveness in both text-to-image generation and editing tasks.

**Questions:**

1. The logic chain from SoftREPA to Soft Tokens is not clearly demonstrated in the paper. Why use Soft Tokens instead of other finetuning techniques like LoRA? Furthermore, LoRA does not put more burden upon time cost by merging them into the model.
Please also refer to Weaknesses part.

**Ethical Concerns:**

["NO or VERY MINOR ethics concerns only"]

**Final Justification:**

The author well clarified the relationship between Soft Tokens and SoftREPA and provided results between LoRA and Soft Tokens. After considering the other reviewers’ remarks, I believe that the inclusion of LoRA only partially diminishes the paper’s overall significance. Accordingly, I have raised my rating to 4.

**Limitations:**

Please refer to Weaknesses part.

**Quality:**

2

**Strengths And Weaknesses:**

Strengths:
1. The proposed SoftREPA is a lightweight contrastive fine-tuning method that introduces fewer than 1M additional trainable parameters, making it highly efficient and easy to integrate with existing pretrained models.
2. The method is backed by a solid theoretical analysis showing that it explicitly increases mutual information between text and image representations, which leads to improved semantic consistency.
3. The approach is evaluated on both text-to-image generation and text-guided image editing tasks, demonstrating consistent improvements in semantic alignment and task performance across multiple benchmarks.

Weaknesses:
1. The paper lacks sufficient comparison between the proposed Soft Tokens and other fine-tuning techniques like LoRA. Otherwise, the authors should provide more explanations on why Soft Tokens are more suitable for the settings.
2. The qualitative and quantitative results seem to be contradictory with each other. For example, according to the Table 1, the performance of SD3 in Counting is better than SD3 + Ours. However, the conclusion is not supported by the results of 'Three zebras' in Figure 3.

---

> ### Author Rebuttal · Authors · 2025-07-31
>
> ### **W1Q1: Comparison with LoRA and Motivation for Soft Tokens**
> We appreciate the reviewer giving us the opportunity to clarify the motivation for soft tokens. The motivation of SoftREPA comes from REPA [1], which shows that aligning the internal representations of DiT with an external pre-trained visual encoder during training significantly improves both discriminative and generative performance. In this paper, we aim to enhance text-image alignment in  **pretrained** text-to-image (T2I) generative models, leveraging these ideas to align DiT’s internal representations through contrastive learning, which is a core principle of representation learning.
>
> The architectural motivation to use soft tokens is primarily inspired by CoOP [2], as stated in our related works. CoOP proposed prompt learning methods for CLIP-like vision-language models that replace the fixed prompt “a photo of a [CLS]” with learned soft prompts that better distinguish between a predefined set of [CLS]. This implies that we can learn an additional feature space that maximizes the discrepancy between general texts, which further enhances alignment with images. While CoOP learns tokens only in the prompt space, we expand this into the feature space within the model and remove the constraint of a predefined class set. This enables more flexible and implicit text-image representation alignment.
>
> To evaluate the effectiveness of our design, we further investigated whether contrastive learning alone could yield similar benefits. For this, we incorporated *LoRA* as an alternative tuning strategy in place of soft tokens. LoRA was applied to the top 5 transformer layers, which we found to be a more stable and effective configuration than applying it to the full model.
>
> As presented in **Table 1**, the combination of *Soft Tokens + Contrastive Loss* consistently achieves stronger performance across key metrics such as *ImageReward*, *CLIP*, *HPS*, and *FID*, while remaining competitive on *PickScore* and *LPIPS*. These results affirm that both soft tokens and contrastive learning contribute independently and meaningfully to enhancing text-image alignment.
>
> **Table 1. Quantitative comparison of parameter-efficient tuning methods**
> | Method                        | ImageReward | PickScore |  CLIP  |  HPS  |  FID  |  LPIPS  |
> |-------------------------------|--------------------|----------------|----------|---------|---------|-----------|
> | SD3                              | 90.89               | 22.50         | 26.26  | 27.97 | **56.87** |**42.47** |
> | LoRA + Contrastive loss | 97.72             | **22.57**    | 26.42  | 28.44 | 70.33 | 42.71 |
> | Soft token + Contrastive loss | **106.32** |22.49 | **26.92**  | **28.70** | 60.76 | 42.99 |
>
> **References**
> [1] ​​Yu, Sihyun, et al. "Representation alignment for generation: Training diffusion transformers is easier than you think." arXiv preprint arXiv:2410.06940 (2024).
> [2] Kaiyang Zhou, Jingkang Yang, Chen Change Loy, and Ziwei Liu. Learning to prompt for vision-language models. International Journal of Computer Vision, 130(9):2337–2348, 2022.
>
> ---
> ### **W2: Qualitative vs Quantitative Results Contradictory**
> We thank the reviewer for pointing out this discrepancy. In Figure 3,  the original motivation to include the example of "Three zebras" was to qualitatively illustrate improvements in **image quality and fidelity**, rather than to make claims about counting ability. While the SD3 baseline struggles to generate complete and realistic zebras, our method produces more coherent and visually accurate outputs, including improved background details. To avoid confusion, we will replace this with other example in the final paper.

---

> ### Author Response · Authors · 2025-08-05
>
> Dear Reviewer 2H9G,
>
> As the deadline for the Author-Reviewer discussion period is approaching, we would like to kindly ask whether our responses have sufficiently addressed your concerns and questions.
>
> We hope our responses have clarified the points you raised, and we are happy to discuss further if you have any remaining questions.
>
> Sincerely,
> The Authors

---

> > ### Comment · Reviewer_2H9G · 2025-08-05
> >
> > Thank you for the detailed reply, which satisfactorily clarified the relationship between Soft Tokens and SoftREPA. Concerning the LoRA configuration, a more rigorous comparison would train the competing methods under essentially identical trainable parameters—either by applying LoRA to all layers or, at minimum, by reporting results for both a “top-5-layers” LoRA and a full-layer LoRA. The rank chosen for each LoRA module should also be disclosed. After considering the other reviewers’ remarks, I believe that the inclusion of LoRA only partially diminishes the overall significance. Accordingly, I would raise my rating to 4.

---

> > > ### Author Response · Authors · 2025-08-06
> > > **Thank you for increasing your score!**
> > >
> > > Thank you very much for your positive feedback and for raising your score. We appreciate your suggestion to perform a more rigorous comparison on LoRA configuration. In response, we report results for both top-5-layer and full-layer LoRA, as well as the corresponding number of trainable parameters and the chosen LoRA rank. Specifically, LoRA with top-5 layers uses 0.4M parameters, LoRA applied to all layers uses 2.3M parameters, and our soft token approach uses 0.9M parameters for 5 layers. We adopt a LoRA rank of 4, which is commonly used for diffusion model finetuning.
> > >
> > > To enable a direct comparison with a similar number of parameters, we additionally report results for LoRA applied to the top-10 layers (0.9M parameters) and top-5 layers with rank 8 (0.9M parameters). Among these settings, LoRA on the top-5 layers with rank 4 yields the best results within LoRA baselines. Importantly, the combination of Soft Tokens and Contrastive Loss consistently achieves the strongest performance across key metrics, supporting our claim that both techniques independently and meaningfully improve text-image alignment. We hope this comprehensive comparison addresses the reviewer’s concerns. The detailed comparison is summarized in **Table 2** below.
> > >
> > > ---
> > > **Table2. Rigorous comparison on LoRA parameters**
> > > | Method                        | Layers | # of Params | ImageReward | PickScore |  CLIP  |  HPS  |  FID  |  LPIPS  |
> > > |---------------------------|:-------------:|:-------------:|:---------------:|:----------:|:---------:|:-------:|:------:|:--------:|
> > > | SD3                              |     -            |        -         |90.89               | 22.50         | 26.26    | 27.97  | 56.87  | 42.47 |
> > > | LoRA + Contrastive loss (r=4)| 1-5  |  0.4M  |97.72              | **22.57**    | 26.42  | 28.44 | 70.33 | 42.71 |
> > > | LoRA + Contrastive loss (r=8)| 1-5  | 0.9M  |95.92              | 22.55   | 26.28  | 28.28 | 70.30 | 42.55 |
> > > | LoRA + Contrastive loss (r=4)| 1-10  | 0.9M  |92.72              | 22.54    | 26.23  | 28.10 | 70.53 | 42.59 |
> > > | LoRA + Contrastive loss (r=4)| all layers  | 2.3M  | 80.89              | 22.20    | 26.70 | 25.97 | 71.57 | **41.54** |
> > > | Soft token + Contrastive loss |   1-5    |  0.9M | **106.32** |22.49 | **26.92**  | **28.70** | **60.76** | 42.99 |

---

### Official Review · Reviewer_1MZz · 2025-07-08

**Clarity:** 2
**Significance:** 1
**Originality:** 2
**Rating:** 3
**Confidence:** 4

**Summary:**

This paper introduces SoftREPA, a novel and lightweight method to significantly improve text-to-image alignment in diffusion models. The authors argue that conventional training is suboptimal for representation alignment and instead propose a contrastive learning framework that trains a small set of learnable "soft tokens" (fewer than 1M parameters) to better distinguish between matching and non-matching text-image pairs, while keeping the base model's weights frozen. This approach, theoretically grounded in maximizing mutual information, is shown to be highly effective and efficient, enhancing prompt fidelity for both image generation and text-guided editing across various models like Stable Diffusion 1.5, XL, and 3, with negligible additional computational overhead.

**Questions:**

1. To better isolate the contribution of the contrastive loss from the soft token architecture, have you considered applying the same loss function with an alternative parameter-efficient tuning method like LoRA?
2. How does it perform on prompts requiring complex compositional understanding, such as intricate spatial relationships (e.g., "a red cube on top of a blue sphere to the left of a green pyramid") or abstract concepts?
3. The paper positions SoftREPA as an alternative to preference-tuning methods like DPO [1] and DRTune [2]. Could these two approaches be complementary?
4. Did the authors explore the impact of different negative sampling strategies, such as using "hard negatives" (e.g., pairing an image with a very similar but incorrect text prompt) versus the current batch-based random sampling?

[1] Wallace B, Dang M, Rafailov R, et al. Diffusion model alignment using direct preference optimization[C]//Proceedings of the IEEE/CVF Conference on Computer Vision and Pattern Recognition. 2024: 8228-8238.
[2] Wu X, Hao Y, Zhang M, et al. Deep reward supervisions for tuning text-to-image diffusion models[C]//European Conference on Computer Vision. Cham: Springer Nature Switzerland, 2024: 108-124.

**Ethical Concerns:**

["NO or VERY MINOR ethics concerns only"]

**Final Justification:**

According to their rebuttal, the results for both components do not significantly outperform the baselines. Consequently, diffusion-DPO + LoRA yields results similar to those in this paper, which greatly diminishes its value. Additionally, it omits a method I mentioned earlier—DRTune—which represents gradient backpropagation instead of DPO that only utilizes value signals.

More importantly, the main issue is that the two methods presented in this paper are isolated and lack a clear connection.

**Limitations:**

Yes.

**Paper Formatting Concerns:**

None.

**Quality:**

2

**Strengths And Weaknesses:**

Strengths:
1. The approach demonstrates significant and broadly generalizable performance gains across various major diffusion models and tasks.
2. Its effectiveness is validated through comprehensive experiments using a wide range of metrics, including human preference scores and detailed ablation studies.
3. The core concepts are presented with exceptional clarity, aided by intuitive figures that simplify complex mechanisms for the reader.

Weaknesses:
1. The main weakness is that the two components are not strongly related. The author should first test fine-tuning with LORA to better illustrate the usefulness of contrastive loss, as the soft tokens are relatively independent.
2. This method may slightly reduce perceptual image quality metrics like FID and LPIPS because it prioritizes stricter adherence to text.
3. It can introduce new failure modes, such as decreased counting accuracy, by over-interpreting alignment cues in text prompts.
4. The title "Easier Than You Think" is somewhat exaggerated, as implementing the method still requires a significant contrastive training setup on large datasets.

---

> ### Author Rebuttal · Authors · 2025-07-31
>
> We sincerely thank the reviewer for their thoughtful and constructive feedback. The comment helped us gain deeper insight and refine the contribution of our work. Below, we address the concern in detail.
>
> ---
> ### **W1/Q1: Isolating the Contribution of Contrastive Loss**
> We appreciate the reviewer’s suggestion to examine the effect of contrastive loss independent of the soft token architecture. In response, we conducted an ablation study using **LoRA** as an alternative in place of soft tokens. Specifically, we applied LoRA to the top 5 transformer blocks, which we found to be more effective than applying LoRA across all layers.
>
> As shown in **Table 1**, the combination of *Soft Tokens + Contrastive Loss* consistently outperforms other variants across metrics such as *ImageReward*, *CLIP*, *HPS*, and *FID*, and performs comparably on *PickScore* and *LPIPS*. This result highlights that both *soft tokens* and *contrastive learning* contribute independently and positively to text-image alignment.
>
> Moreover, these findings support the underlying motivation of SoftREPA: to explore the text feature space to better text-image representation alignment. While LoRA modifies the attention projection layers in both the text and image encoders, this direct parameter modification may lead to suboptimal image generation, as evidenced by a higher *FID* score (70.33) compared to our soft token training approach (60.76).
>
> **Table 1. Quantitative comparison of parameter-efficient tuning methods**
> | Method                         | ImageReward | PickScore |  CLIP  |  HPS  |  FID  |  LPIPS  |
> |-------------------------------|--------------|-----------|--------|-------|--------|---------|
> | SD3 (base)                    | 90.89        | 22.50     | 26.26  | 27.97 | **56.87**  | **42.47**   |
> | LoRA + Contrastive Loss       | 97.72        | **22.57** | 26.42  | 28.44 | 70.33  | 42.71   |
> | Soft Token + Contrastive Loss | **106.32**| 22.49|**26.92**|**28.70**| 60.76 | 42.99   |
>
> ---
> ### **W2: Drop in FID and LPIPS**
> We acknowledge that our method may affect FID and LPIPS, as it emphasizes stricter text-image alignment. However, these metrics have limitations in evaluating generation quality alone—FID is influenced by both diversity and quality, while LPIPS is more suited to measure similarity between a real image and a generated image on the given text, rather than assessing image quality.
>
> Instead,  other metrics such as ImageReward, PickScore, and HPS  reflect the image quality aligned with human preferences. These metrics consistently demonstrate that our method maintains perceptual quality while improving text-image alignment.
>
> ---
> ### **W3Q4: Counting Accuracy and Hard Negatives**
> We thank the reviewer for suggesting alternative approaches to improve counting accuracy. To address this, we trained SoftREPA with hard negatives added to each sample. Specifically, for each COCO caption, we generated three hard negative examples focusing on the following aspects: object count, color, and position. Using GPT-4.1-mini, we crafted prompts to minimally modify the original caption by altering the number of objects (counting), the color of an object (color), or the position of an object (position). During training, for each training sample, these hard negatives were incorporated alongside other negative samples.
>
> As shown in ***Table 2***, we did not observe a significant improvement from adding hard negatives to the training data. This may be due to the limited number of hard negatives, as we generated only one altered caption per attribute. We hope to conduct further experiments on the impact of hard negatives in future work.
>
> **Table 2: Effect of Incorporating Hard Negatives into Our Method**
> | Method                        | Mean | Single | Two | Counting | Colors | Position | Color Attr.|
> |-------------------------------|----------|--------|----------|-----------|-----------|------------|---------------|
> | SD3                              | 0.68  |  0.99  | 0.86    | **0.56**     | 0.85     |   0.27     |    0.55      |
> | Ours                            | **0.70**  |  **1.00**   | **0.95**    | 0.29     | **0.92**     |  **0.34**     |   **0.68**      |
> | Ours+Hardneg          | 0.68  |  **1.00**   | 0.86    | **0.56**     | 0.84     |   0.28     |   0.53      |
>
>
>
> ---
> ### **W4: Title May Be Overstated**
> We will tone down and  revise it to better reflect the scope of our method.
>
> ---
> ### **Q2: Complex Compositional Understanding**
> We thank the reviewer for this important question regarding compositional understanding. We address this capability primarily in the Editing tasks. To evaluate this aspect, we used LLaVA [1] to generate long, detailed text prompts for images in the DIV2K dataset. These prompts contain nuanced edits involving spatial relationships, attribute modifications, and object substitutions, requiring complex compositional reasoning.
>
> Below, we present two representative prompts we used.  In each case, the target prompt modifies multiple attributes and compositional elements from the original source prompt, such as changing object positions, attributes (e.g., material, color), or introducing new objects in specific spatial contexts.
>
> In the **DIV2K Editing** benchmark, our model with trained soft tokens consistently outperforms the base model on metrics that evaluate background preservation and alignment with the target prompt (refer to Table 2 in the paper). These results suggest that the incorporation of soft tokens enhances the model’s ability to reason over complex, multi-object compositions and produce more faithful edits.
>
> ---
> **Example 1**: The image features a vase filled with a variety of flowers, including pink, white, and red ones. The vase is placed on **{a table, and it is a beautiful blue color $\rightarrow$  a wooden pedestal, and it is a rich, dark brown color**}. The flowers are arranged in a bouquet, creating an eye-catching display. The combination of the vibrant colors and the elegant *blue vase* makes the scene visually appealing.
>
> **Example 2**: The image features a room with a wooden wall and a wooden table. On the wall, there are several framed pictures, including a picture of **{a man wearing a suit $\rightarrow$ a woman wearing a dress}**. The table is covered with a variety of items, such as books, a vase, and a bottle. There is also a **{jacket hanging $\rightarrow$ a guitar hanging}** on the wall, adding to the room's decor. The room appears to be a combination of a living space and a workspace, with the wooden wall and table providing a cozy and functional atmosphere.
>
> ---
> **References**
> [1] Haotian Liu, Chunyuan Li, Qingyang Wu, and Yong Jae Lee. Visual instruction tuning. Advances in neural information processing systems, 36:34892–34916, 2023.
>
> ---
> ### **Q3: Diffusion RL + SoftREPA Complementarity**
> We thank the reviewer for this insightful question regarding the potential complementarity between SoftREPA and existing preference-tuning methods. We agree that these approaches are not mutually exclusive, and we explored this direction with additional experiments. To investigate whether SoftREPA can be combined with diffusion-RL methods, we conducted experiments using two representative Diffusion RL frameworks: Diffusion-DPO [1], which performs preference optimization, and DDPO [2], which utilizes policy optimization based on an external reward (e.g., BERTScore [3]).
>
> To test complementarity, we adopted a **plug-and-play** approach: we integrated pretrained soft tokens from SoftREPA into the inference pipelines of these diffusion-RL models **without further training**.
>
> As shown in **Table 3**, the addition of SoftREPA's soft tokens leads to notable performance gains in both cases. *Diffusion-DPO + Soft Tokens* shows improvements in *ImageReward*, *CLIP*, *FID*, and *LPIPS*, and remains comparable in *PickScore* and *HPS*. *DDPO + Soft Tokens* yields consistent improvements across all metrics, indicating strong synergy between policy-based optimization and our contrastive alignment objective.
>
> These results suggest that SoftREPA and Diffusion-RL methods are complementary, while Diffusion-DPO and DDPO aim to align outputs using preference or reward supervision, SoftREPA enhances internal representation alignment through contrastive learning. Their combination allows for leveraging strengths from both strategies—efficient text-image alignment and explicit preference feedback—to further improve generation quality.
>
> **Table 3. Complementarity with Diffusion RL methods**
> | Method                                   | ImageReward | PickScore |  CLIP  |  HPS  |  FID  |  LPIPS  |
> |----------------------------------------|--------------------|----------------|----------|---------|---------|-----------|
> | Diffusion-DPO[1]                    | 22.26          | **21.68** | 26.51  | **25.60** | 54.19 | 43.96    |
> | Diffusion-DPO[1] + Ours        | **33.10**  | 21.64 | **27.41**  | 25.55 | **53.72** | **43.64** |
> | DDPO[2]                                 | -8.83               | 21.20         | 26.21  | 23.45 | 54.83 | 43.85 |
> | DDPO[2] + Ours                     | **8.07** |**21.25**|**27.13**|**23.75**|**54.18**| **43.59** |
>
> **References**
> [1] Wallace, Bram, et al. "Diffusion model alignment using direct preference optimization." Proceedings of the IEEE/CVF Conference on Computer Vision and Pattern Recognition. 2024.
> [2] Black, Kevin, et al. "Training diffusion models with reinforcement learning." arXiv preprint arXiv:2305.13301 (2023).
> [3] Zhang, Tianyi, et al. "Bertscore: Evaluating text generation with bert." arXiv preprint arXiv:1904.09675 (2019).

---

> ### Author Response · Authors · 2025-08-05
>
> Dear Reviewer 1MZz,
>
> As the deadline for the Author-Reviewer discussion period is approaching, we would like to kindly ask whether our responses have sufficiently addressed your concerns and questions.
>
> We hope our responses have clarified the points you raised, and we are happy to discuss further if you have any remaining questions.
>
> Sincerely,
> The Authors

---

> > ### Author Response · Authors · 2025-08-08
> >
> > Dear Reviewer 1MZz,
> >
> > We sincerely appreciate the time you’ve taken to review our work. As the discussion phase is drawing to a close, we wanted to confirm that our earlier responses have resolved your concerns.  Please let us know if any issues remain; we will gladly clarify them.

---

### Note · Authors · 2025-08-12

Dear Area Chair and Reviewers,

We sincerely appreciate the insightful and constructive discussion period, which has helped us strengthen and clarify our paper. In response to feedback, we have addressed the reviewers' concerns by incorporating substantial new experiments and analyses:

**General Review - Validated Contributions of Soft Tokens and Contrastive Loss:**
Through targeted ablations—replacing soft tokens with LoRA parameters and contrastive loss with DSM loss—we confirmed that **SoftREPA’s synergy of these components consistently improves text-image alignment** from both structural and training perspectives.

**Reviewer 1MZz - Complementarity with Diffusion RL:**
Thanks to the reviewer’s suggestion, we showed that **SoftREPA-trained soft tokens integrate plug-and-play** with RL-finetuned models (Diffusion-DPO, DDPO) without retraining. We show potential to open a new, orthogonal axis of performance improvement for human preference alignment and text-image fidelity.  We hope that these updates are helpful as you finalize your review.

**Reviewer 2H9G - Rigorous Comparison with LoRA Configurations:**
To further address the reviewer’s concerns from the discussion period regarding the relationship between soft tokens and our method, we compared SoftREPA to LoRA finetuning across equivalent parameter counts, layer selections, and LoRA ranks. These experiments demonstrate the significance of our approach under identical parameter budgets.

**Reviewers hkeq, v3Q6 - Comparison with Related Works:**
We extended comparisons to training-free and RL-based finetuning methods on the GenEval benchmark. Even against full-finetuning RL approaches, our lightweight finetuning strategy achieves **competitive or superior performance**.

**Reviewer F1wd - Generalization Across Models:**
We demonstrated strong generalization across SD1.5, SDXL, and SD3, with and without DSM loss. Additionally, we show consistent hyperparameter trends—highlighting **ease of adoption** across architectures.  We are pleased to hear that your original concerns have been properly addressed.

We believe these revisions directly address all major reviewer concerns and substantially strengthen the paper.  SoftREPA emerges as a novel, lightweight, and robust framework for improving text-image alignment in diffusion models, with broad applicability and complementarity to existing methods. We are confident it will serve as a valuable contribution to the community.

Thank you

---

### Decision · Program_Chairs · 2025-09-17

**Decision:**

Accept (poster)

**Comment:**

This paper introduces SoftREPA, a novel method to improve text-to-image alignment in diffusion models via a contrastive learning setup that trains a set of learnable soft tokens. The approach is theoretically well grounded and can work effectively across multiple existing models. Authors have clarified most of the questions raised by the reviewers, in particular comparison of the method against LoRA. I recommend acceptance for this work.